# Residual Stream Analysis with Multi-Layer SAEs

**Tim Lawson*** **Lucy Farnik** **Conor Houghton** **Laurence Aitchison**
School of Engineering Mathematics and Technology
University of Bristol
Bristol, UK

## Abstract

Sparse autoencoders (SAEs) are a promising approach to interpreting the internal representations of transformer language models. However, SAEs are usually trained separately on each transformer layer, making it difficult to use them to study how information flows across layers. To solve this problem, we introduce the multi-layer SAE (MLSAE): a single SAE trained on the residual stream activation vectors from every transformer layer. Given that the residual stream is understood to preserve information across layers, we expected MLSAE latents to 'switch on' at a token position and remain active at later layers. Interestingly, we find that individual latents are often active at a single layer for a given token or prompt, but the layer at which an individual latent is active may differ for different tokens or prompts. We quantify these phenomena by defining a distribution over layers and considering its variance. We find that the variance of the distributions of latent activations over layers is about two orders of magnitude greater when aggregating over tokens compared with a single token. For larger underlying models, the degree to which latents are active at multiple layers increases, which is consistent with the fact that the residual stream activation vectors at adjacent layers become more similar. Finally, we relax the assumption that the residual stream basis is the same at every layer by applying pre-trained tuned-lens transformations, but our findings remain qualitatively similar. Our results represent a new approach to understanding how representations change as they flow through transformers. We release our code to train and analyze MLSAEs at `https://github.com/tim-lawson/mlsae`.

## 1 Introduction

Sparse autoencoders (SAEs) learn interpretable directions or 'features' in the representation spaces of language models (Elhage et al., 2022; Cunningham et al., 2023; Bricken et al., 2023). Typically, SAEs are trained on the activation vectors from a single model layer (Gao et al., 2024; Templeton et al., 2024; Lieberum et al., 2024). This approach illuminates the representations within a layer. However, Olah (2024); Templeton et al. (2024) believe that models may encode meaningful concepts by simultaneous activations in multiple layers, which SAEs trained at a single layer do not address. Furthermore, it is not straightforward to automatically identify correspondences between features from SAEs trained at different layers, which may complicate circuit analysis (e.g. He et al., 2024).

To solve this problem, we take inspiration from the residual stream perspective, which states that transformers (Vaswani et al., 2017) selectively write information to and read information from token positions with self-attention and MLP layers (Elhage et al., 2021; Ferrando et al., 2024). The results of subsequent circuit analyses, like the explanation of the indirect object identification task presented by Wang et al. (2022), support this viewpoint and cause us to expect the activation vectors at adjacent layers in the residual stream to be relatively similar (Lad et al., 2024).

To capture the structure shared between layers in the residual stream, we introduce the multi-layer SAE (MLSAE): a single SAE trained on the residual stream activation vectors from every layer of a transformer language model. Importantly, the autoencoder itself has a single hidden layer – it is

---

*Correspondence to `tim.lawson@bristol.ac.uk`.

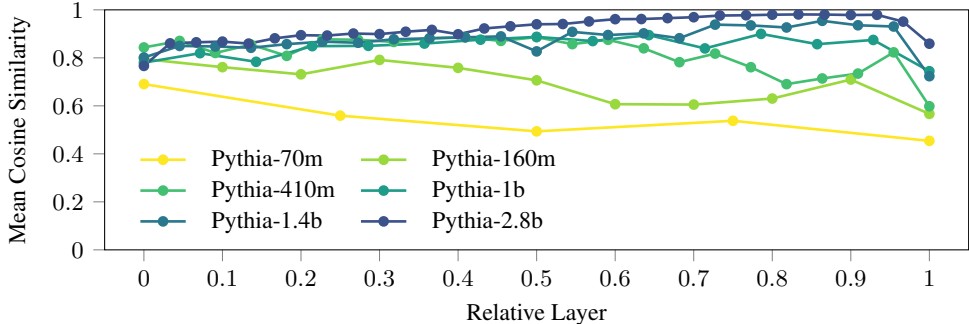

Figure 1: The mean cosine similarities between the residual stream activation vectors at adjacent layers of transformers, over 10 million tokens from the test set. To compare transformers with different numbers of layers, we divide the lower of each pair of adjacent layers by the number of pairs. This 'relative layer' is the $x$-axis of the plot. We subtract the dataset mean from the activation vectors at each layer before computing cosine similarities to control for changes in the norm between layers (Heimersheim & Turner, 2023), which we demonstrate in Figure 4.

multi-layer only in the sense that it is trained on activations from multiple layers of the underlying transformer. In particular, we consider the activation vectors from each layer as separate training examples, which is equivalent to training a single SAE at each layer individually but with the parameters tied across layers. We briefly discuss alternative methods in Section 5.

We show that multi-layer SAEs achieve comparable reconstruction error and downstream loss to single-layer SAEs while allowing us to directly identify and analyze features that are active at multiple layers (Section 4.1). When aggregating over a large sample of tokens, we find that individual latents are likely to be active at multiple layers, and this measure increases with the number of latents. However, for a single token, latent activations are more likely to be isolated to a single layer. For larger underlying transformers, we show that the residual stream activation vectors at adjacent layers are more similar and that the degree to which latents are active at multiple layers increases.

Finally, we relax the assumption that the residual stream basis is the same at every layer by applying pre-trained tuned-lens transformations to activation vectors before passing them to the encoder. Surprisingly, this does not obviously increase the extent of multi-layer latent activations.

## 2 RELATED WORK

A sparse code represents many signals, such as sensory inputs, by simultaneously activating a relatively small number of elements, such as neurons (Olshausen & Field, 1996; Bell & Sejnowski, 1997). Sparse dictionary learning (SDL) approximates each input vector by a linear combination of a relatively small number of learned basis vectors. The learned basis is usually overcomplete: it has a greater dimension than the inputs. Independent Component Analysis (ICA) achieves this aim by maximizing the statistical independence of the learned basis vectors by iterative optimization or training (Bell & Sejnowski, 1995; 1997; Hyvärinen & Oja, 2000; Le et al., 2011). Sparse autoencoders (SAEs) can be understood as ICA with the addition of a noise model optimized by gradient descent (Lee et al., 2006; Ng, 2011; Makhzani & Frey, 2014)

The activations of language models have been hypothesized to be a dense, compressed version of a sparse, expanded representation space (Elhage et al., 2021; 2022). Under this view, there are interpretable directions in the dense representation spaces corresponding to distinct semantic concepts, whereas their basis vectors (neurons) are 'polysemantic' (Park et al., 2023). It has been shown theoretically (Wright & Ma, 2022) and empirically (Elhage et al., 2022; Sharkey et al., 2022; Whittington et al., 2023) that SDL recovers ground-truth features in toy models, and that learned dictionary elements are more interpretable than the basis vectors of language models (Cunningham et al., 2023; Bricken et al., 2023) or dense embeddings (O'Neill et al., 2024). Notably, 'features' are not necessarily linear (Wattenberg & Viégas, 2024; Engels et al., 2024; Hernandez et al., 2024).

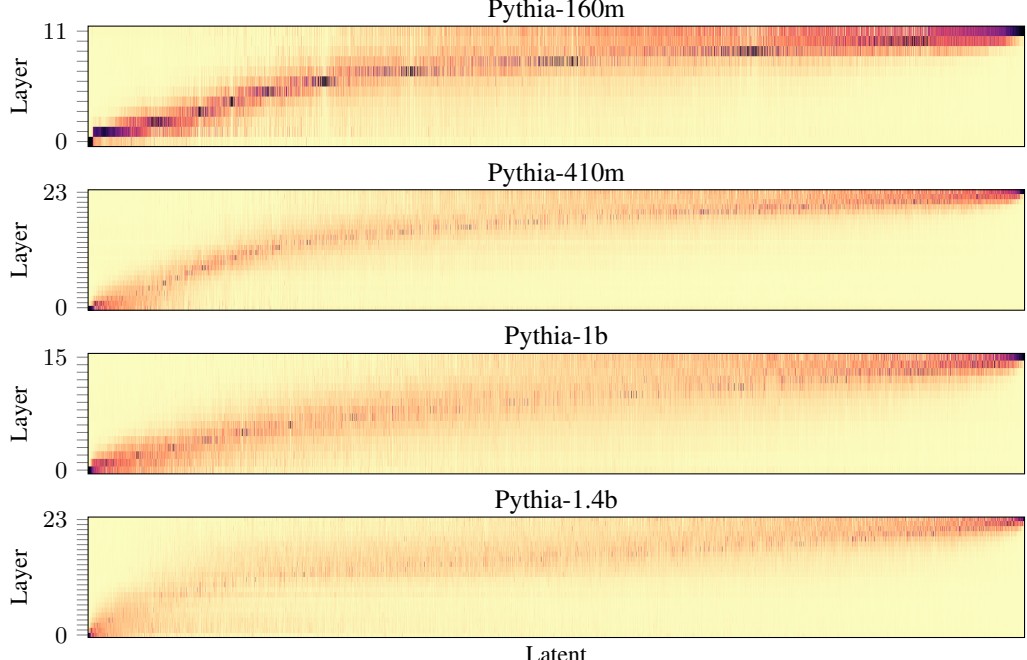

Figure 2: Heatmaps of the distributions of latent activations over layers when aggregating over 10 million tokens from the test set. Here, we plot the distributions for MLSAEs trained on Pythia models with an expansion factor of $R = 64$ and sparsity $k = 32$. The latents are sorted in ascending order of the expected value of the layer index (Eq. 10).

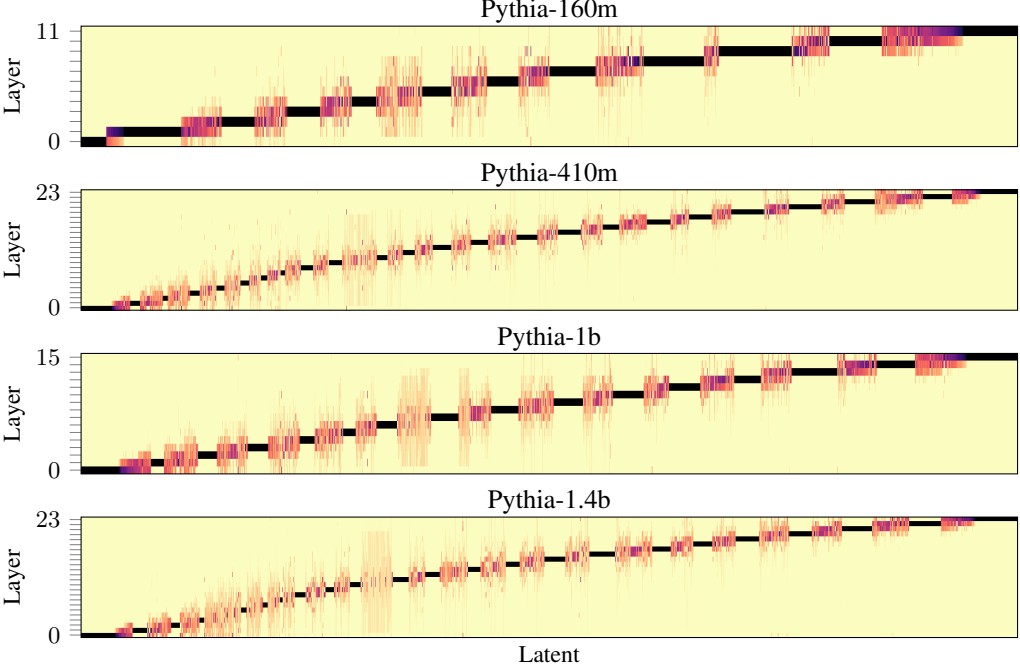

Figure 3: Heatmaps of the distributions of latent activations over layers for a single example prompt. Here, we plot the distributions for MLSAEs trained on Pythia models with an expansion factor of $R = 64$ and sparsity $k = 32$. The example prompt is "When John and Mary went to the store, John gave" (Wang et al., 2022). We exclude latents with maximum activation below $1 \times 10^{-3}$ and sort latents in ascending order of the expected value of the layer index (Eq. 10).

The standard SAE architecture is a single hidden layer with a ReLU activation function and an $L^1$ sparsity penalty in the training loss (Bricken et al., 2023), but various activation functions (Makhzani & Frey, 2014; Konda et al., 2015; Rajamanoharan et al., 2024b;a) and objectives (Braun et al., 2024) have been proposed. The prevailing approach is to train an SAE on the activation vectors from a single transformer layer, except for Kissane et al. (2024), who concatenate the outputs of multiple attention heads in a single layer, and Yun et al. (2021), who learn a sparse dictionary for the residual stream at multiple layers, albeit by iterative optimization instead of with an autoencoder.

Mechanistic interpretability research often attempts to identify circuits: computational subgraphs of neural networks that implement specific behaviors (Olah et al., 2020; Wang et al., 2022; Conmy et al., 2023; Dunefsky et al., 2024; García-Carrasco et al., 2024; Marks et al., 2024). Representing networks in terms of SAE latents may help to improve circuit discovery (He et al., 2024; O'Neill & Bui, 2024), and these latents can be used to construct steering vectors (Subramani et al., 2022; Templeton et al., 2024; Makelov, 2024), but it is unclear whether SAEs outperform baselines for causal analysis (Chaudhary & Geiger, 2024; Huang et al., 2024). Importantly, SAEs can be scaled up to the activations of large language models, where we expect the number of distinct semantic concepts to be extremely large (Templeton et al., 2024; Gao et al., 2024; Lieberum et al., 2024).

The 'logit lens' is a method to interpret directions in the residual stream by projecting them onto the vocabulary space to elicit token predictions, i.e., multiplying them by the unembedding matrix (nostalgebraist, 2020). However, the residual stream basis is not fixed, so Belrose et al. (2023) introduce the 'tuned lens' approach, where a linear transformation is learned for each layer in the residual stream. The objective is to minimize the KL divergence between the probability distribution over tokens generated by the transformed activations and the 'true' distribution of the model. This approach draws on the perspective of iterative inference (Jastrzębski et al., 2018).

The key difference between previous work (Bricken et al., 2023; Cunningham et al., 2023; Templeton et al., 2024; Gao et al., 2024) and our work is that we introduce the multi-layer SAE, i.e., we train a single SAE at all layers of the residual stream.

## 3 METHODS

The key idea with a multi-layer SAE is to train a single SAE on the residual stream activation vectors from every layer. In particular, we consider the activations at each layer to be different training examples. Hence, for residual stream activation vectors of model dimension $d$, the inputs to the multi-layer SAE also have dimension $d$. For $n_T$ tokens and $n_L$ layers, we train the multi-layer SAE on $n_T n_L$ activation vectors. We use the terms 'SAE feature' and 'latent' interchangeably.

### 3.1 SETUP

We train MLSAEs primarily on GPT-style language models from the Pythia suite (Biderman et al., 2023); see Appendix B.3 for others. We are interested in the computation performed by self-attention and MLP layers on intermediate representations (Valeriani et al., 2023). Hence, we take the residual stream activation vectors after a given transformer block has been applied, excluding the input embeddings before the first block and taking the last-layer activations before the final layer norm.

We use a $k$-sparse autoencoder (Makhzani & Frey, 2014; Gao et al., 2024), which directly controls the sparsity of the latent space by introducing a TopK activation function that keeps only the $k$ largest latents. The $k$ largest latents are almost always positive for $k \ll d$, but we follow Gao et al. (2024) in applying a ReLU activation function to guarantee non-negativity. This setup effectively fixes the sparsity ($L^0$ norm) of the latents at $k$ per activation vector (layer and token) throughout training. For input vectors $\mathbf{x} \in \mathbb{R}^d$ and latent vectors $\mathbf{h} \in \mathbb{R}^n$, the encoder and decoder are defined by:

$$\mathbf{h} = \mathrm{ReLU}(\mathrm{TopK}(\mathbf{W}_{\mathrm{enc}}\mathbf{x} - \mathbf{b}_{\mathrm{pre}})) \tag{1}$$

$$\hat{\mathbf{x}} = \mathbf{W}_{\mathrm{dec}}\mathbf{h} + \mathbf{b}_{\mathrm{pre}} \tag{2}$$

where $\mathbf{W}_{\mathrm{enc}} \in \mathbb{R}^{d \times n}$, $\mathbf{W}_{\mathrm{dec}} \in \mathbb{R}^{n \times d}$, and $\mathbf{b}_{\mathrm{pre}} \in \mathbb{R}^d$. We constrain the pre-encoder bias $\mathbf{b}_{\mathrm{pre}}$ to be the negative of the post-decoder bias, following Bricken et al. (2023); Gao et al. (2024), and standardize activation vectors to zero mean and unit variance before passing them to the encoder.

## 3.2 TRAINING

We use the fraction of variance unexplained (FVU) as the reconstruction error:

$$\text{FVU}(\mathbf{x}, \hat{\mathbf{x}}) = \frac{\|\mathbf{x} - \hat{\mathbf{x}}\|_2^2}{\text{Var}(\mathbf{x})} \tag{3}$$

Here, $\text{Var}$ is the variance. We chose the FVU because the input vectors from different layers may have different magnitudes; choosing the mean squared error (MSE) would encourage the autoencoder to prioritize minimizing the reconstruction errors of the layers with the greatest magnitudes.

A potential issue when training SAEs is the occurrence of 'dead' latents, i.e., latent dimensions that are almost always zero. With a $k$-sparse autoencoder, this means latent dimensions that almost never appear among the $k$ largest latent activations. We follow Bricken et al. (2023); Cunningham et al. (2023) by considering a latent 'dead' if it is not activated within the last 10 million tokens during training. In the multi-layer setting, a latent may be activated by the input vectors from any layer.

Gao et al. (2024, Appendix A.2) propose an auxiliary loss term to minimize the occurrence of dead latents. This AuxK term models the MSE reconstruction error using the $k_\text{aux}$ largest dead latents:

$$\text{AuxK}(\mathbf{x}, \hat{\mathbf{x}}) = \|\mathbf{e} - \hat{\mathbf{e}}\|_2^2 \tag{4}$$

Here, $\mathbf{e} = \mathbf{x} - \hat{\mathbf{x}}$ is the reconstruction error of the main model, and $\hat{\mathbf{e}}$ is its reconstruction using the top-$k_\text{aux}$ dead latents. Let $\text{Dead}$ be an 'activation function' that keeps only the dead latents. Then:

$$\mathbf{h}_\text{dead} = \text{ReLU}(\text{TopK}_\text{aux}(\text{Dead}(\mathbf{W}_\text{enc}\mathbf{x} - \mathbf{b}_\text{pre}))) \tag{5}$$
$$\hat{\mathbf{e}} = \mathbf{W}_\text{dec}\mathbf{h}_\text{dead} + \mathbf{b}_\text{pre} \tag{6}$$

The full loss is the FVU plus the auxiliary loss term, multiplied by a small coefficient $\alpha$:

$$\mathcal{L} = \text{FVU}(\mathbf{x}, \hat{\mathbf{x}}) + \alpha \cdot \text{AuxK}(\mathbf{x}, \hat{\mathbf{x}}) \tag{7}$$

Following Gao et al. (2024), we choose $k_\text{aux}$ as a power of 2 close to $d/2$ and $\alpha = 1/32$.

Our hyperparameters are the expansion factor $R = n/d$, the ratio of the number of latents to the model dimension, and the sparsity $k$, the number of largest latents to keep in the TopK activation function. We choose expansion factors as powers of 2 between 1 and 256, yielding autoencoders with between 512 and 131072 latents for Pythia-70m, and $k$ as powers of 2 between 16 and 512 (Appendix B).

The computational expense of training a single multi-layer SAE on $n_L$ layers of the residual stream is approximately equal to training $n_L$ single-layer SAEs on the same number of tokens. We trained most MLSAEs on a single NVIDIA GeForce RTX 3090 GPU for between 12 and 24 hours; we trained the largest MLSAEs (e.g., with Pythia-1b or an expansion factor of $R = 256$) on a single NVIDIA A100 80GB GPU for up to three days.

The implementation is based on Gao et al. (2023); Belrose (2024); see Appendix A for details.

## 3.3 TUNED LENS

In the tuned lens method, an affine transformation is learned from the output space of layer $\ell$ to the output space of the final layer, called the translator for layer $\ell$ (Belrose et al., 2023). With our setup, we want to transform the residual stream activation vectors at each layer into more similar bases before passing them to the encoder and invert that transformation after the decoder.

Importantly, the authors note that their implementation[1] uses a residual connection:

$$\mathbf{x}' = \mathbf{x} + (\mathbf{W}_{\text{lens},\ell}\mathbf{x} + \mathbf{b}_{\text{lens},\ell}) \tag{8}$$

Here, $\mathbf{x}$ is the input vector to the encoder, and $\mathbf{x}'$ is the transformed input vector. This parameterization ensures that $L_2$ regularization (weight decay) pushes the transformation towards the identity matrix instead of zero. Hence, to invert the transformation, we need:

$$\hat{\mathbf{x}} = (\mathbf{I} + \mathbf{W}_{\text{lens},\ell})^{-1}(\hat{\mathbf{x}}' - \mathbf{b}_{\text{lens},\ell}) \tag{9}$$

---

[1]`https://github.com/AlignmentResearch/tuned-lens`, file: `tuned_lens/nn/lenses.py`

| Model | FVU | $L^1$ Norm | Loss ↑ |
|---|---|---|---|
| Pythia-70m | 0.097 | 66.5 | 0.565 |
| Pythia-160m | 0.106 | 76.1 | 0.432 |
| Pythia-410m | 0.081 | 84.6 | 0.414 |
| Pythia-1b | 0.095 | 109.7 | 0.404 |
| Pythia-1.4b | 0.106 | 124.7 | 0.487 |
| Gemma 2 2B | 0.210 | 255.5 | 1.483 |
| Llama 3.2 3B | 0.299 | 141.4 | 0.347 |
| GPT-2 small | 0.093 | 196.9 | 0.759 |

(a) Standard

| Model | FVU | $L^1$ Norm | Loss ↑ |
|---|---|---|---|
| Pythia-70m | 0.030 | 61.5 | 0.274 |
| Pythia-160m | 0.088 | 89.6 | −0.080 |
| Pythia-410m | 0.073 | 80.5 | 0.827 |

(b) With tuned lens

Table 1: The mean FVU, $L^1$ norm, and increase in the cross-entropy loss (Loss ↑) for MLSAEs with an expansion factor of $R = 64$ and sparsity $k = 32$, over 1 million tokens from the test set. We provide details of the evaluation metrics and transformer architectures in Appendix B.

In Eq. 9, $\hat{\mathbf{x}}'$ is the transformed output vector of the decoder, $\hat{\mathbf{x}}$ is the output vector, $\mathbf{W}_{\text{lens},\ell} \in \mathbb{R}^{d \times d}$, and $\mathbf{b}_{\text{lens},\ell} \in \mathbb{R}^d$. With our setup, $\mathbf{x}'$ and $\hat{\mathbf{x}}'$ replace the input and output vectors that we pass to the encoder and use to compute the loss. Notably, we use the transformed vectors to compute reconstruction errors (Figure 12), and we compute the inverse $(\mathbf{I} + \mathbf{W}_{\text{lens},\ell})^{-1}$ for each layer once at the start of training. The pre-trained tuned lenses used were provided by the authors of Belrose et al. (2023), which did not include Pythia-1b at the time of writing.[2]

## 4 RESULTS

### 4.1 EVALUATION

The key advantage of a multi-layer SAE is to be able to study how information flows across layers in the residual stream. However, this approach is only useful if the MLSAE performs comparably to single-layer SAEs. The FVU reconstruction error term in the loss (Section 3.2) is a proxy for the degree to which an SAE explains the behavior of the underlying model. Hence, we also measured the increase in the cross-entropy loss when the residual stream activations at a given layer are replaced by their reconstruction, following Braun et al. (2024); Gao et al. (2024); Lieberum et al. (2024).

Table 1 summarizes the evaluation results for MLSAEs trained with our default hyperparameters. The FVU and delta cross-entropy (CE) loss remain consistent across model sizes for Pythia transformers. In most cases, applying tuned-lens transformations decreases the FVU and delta CE loss (see Section 4.4 and Figure 12). We provide results for other hyperparameters, breakdowns by the layer of the input activation vectors, and explicit comparisons to single-layer SAEs in Appendix B.

### 4.2 REPRESENTATION DRIFT

Guided by the residual stream perspective (Elhage et al., 2021; Ferrando et al., 2024), we expected dense activation vectors to be relatively similar across layers. As an approximate measure of the degree to which information is preserved in the residual stream, we computed the cosine similarities between the activation vectors at adjacent layers, similarly to Lad et al. (2024, Appendix A). A similarity of one means that the information represented at a token position is unchanged by the intervening residual block, whereas a similarity of zero means the activation vectors on either side of the block are orthogonal. We had expected changes in the residual stream to become smaller as the model size increased, and the mean cosine similarities increased as expected (Figure 1).

Given that the residual stream activation vectors are relatively similar between adjacent layers, we expected to find many MLSAE latents active at multiple layers. We confirmed this prediction over a large sample of 10 million tokens from the test set (Figure 2). Interestingly, we found that for individual prompts, a much greater proportion of latents are active at only a single layer (Figure 3).

---

[2]https://huggingface.co/spaces/AlignmentResearch/tuned-lens

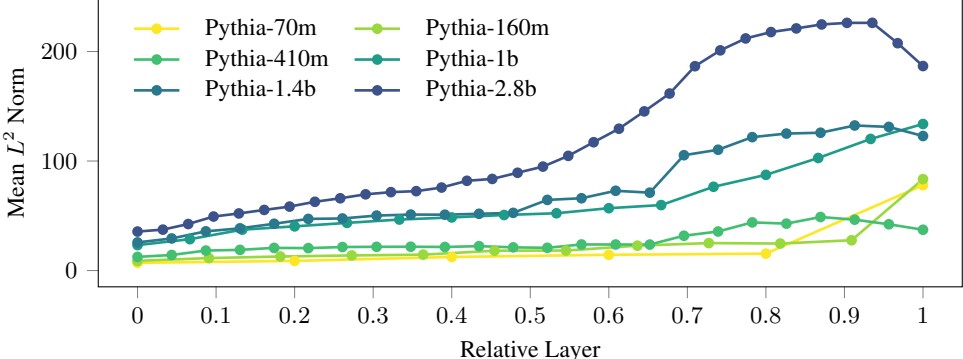

Figure 4: The mean $L^2$ norm of the residual stream activation vectors at every layer, over 10 million tokens from the test set. To compare transformers with different numbers of layers, we divide the layer index $\ell$ by the number of layers $n_L$. This 'relative layer' is the $x$-axis of the plot.

Following Heimersheim & Turner (2023), we verified that the mean $L^2$ norm of the activation vectors increases across layers, which prompted us to center the vectors at each layer by subtracting the dataset mean before computing the similarities between vectors (Figure 4).

### 4.3 LATENT DISTRIBUTIONS OVER LAYERS

Given a dataset and MLSAE, each combination of a token and latent produces a distribution of activations over layers. We want to understand the degree to which the variance of that distribution depends on the token versus the latent to quantify the intuition gleaned from Figures 2 and 3.

Consider the layer index $L$, token $T$, and latent index $J$ to be random variables. We take $P(J)$ to be a uniform discrete distribution, $P(T \mid J)$ to be a uniform discrete distribution over tokens for which the latent is active (at any layer), and $L$ to be sampled from a conditional distribution proportional to the total latent activation at that layer, aggregating over tokens:

$$P(L = \ell \mid T = t,\, J = j) = \frac{h_j(\mathbf{x}_{t,\ell})}{\sum_{\ell'} h_j(\mathbf{x}_{t,\ell'})} \tag{10}$$

Here, $\mathbf{x}_{t,\ell}$ is the dense residual stream activation vector at token $t$ and layer $\ell$, while $h_j(\mathbf{x}_{t,\ell})$ is the activation of the $j$-th MLSAE latent at that token and layer.

We order latents in all heatmaps using the expected value of the layer index for a single latent $\mathbb{E}[L \mid J = j]$. The variance of the distribution over layers measures the degree to which a latent is active at a single layer (in which case, it is zero) versus multiple layers (in which case, it is positive). We are interested in the following variances of the distribution over layers:

- $\mathrm{Var}[L \mid J = j,\, T = t]$, for a single latent and token
- $\mathrm{Var}[L \mid J = j]$, for a single latent, aggregating over tokens
- $\mathrm{Var}[L]$, aggregating over both latents and tokens

These quantities are related by the law of total variance (see Appendix C.2). For the moment, we note that the variance of the distribution over layers naturally depends on the number of layers $n_L$. Hence, to compare different models, we look at ratios between these variances:

$$\begin{array}{r} \text{Variance for one latent, aggregating over tokens,} \\ \text{as a proportion of the total variance over all latents} \end{array} = \frac{\mathbb{E}[\mathrm{Var}(L \mid J)]}{\mathrm{Var}(L)} \tag{11}$$

$$\begin{array}{r} \text{Variance for one token and latent as a} \\ \text{proportion of the total variance for that latent} \end{array} = \frac{\mathbb{E}[\mathrm{Var}(L \mid J,\, T)]}{\mathbb{E}[\mathrm{Var}(L \mid J)]} \tag{12}$$

The former measures the degree to which latents are active at multiple layers when aggregating over tokens, and the latter compares this to the case for a single token. We explore alternative measures of aggregate multi-layer activity in Appendix C.

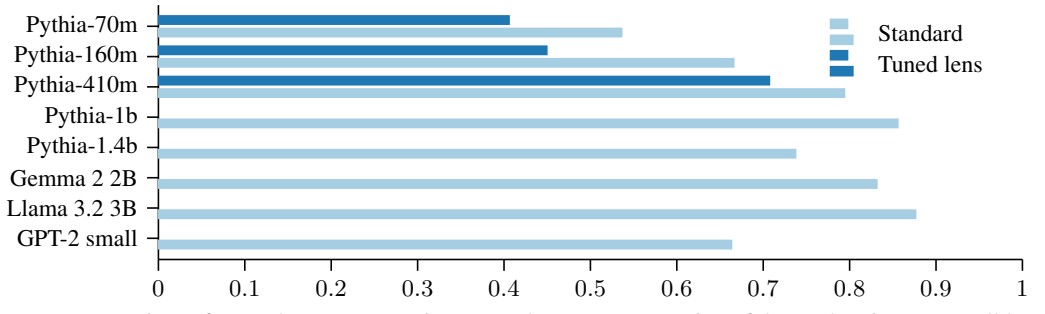

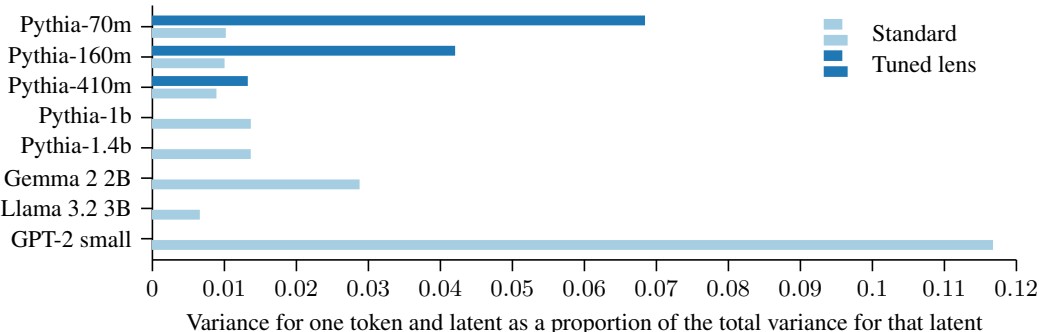

Figure 5: The fraction of the total variance explained by individual latents and the fraction of the variance for an individual latent explained by individual tokens (Eqs. 11 and 12) for MLSAEs with an expansion factor of $R = 64$ and sparsity $k = 32$, over 10 million tokens from the test set. The absence of bars for tuned-lens MLSAEs indicates the absence of results, not that the values are zero.

The degree to which latents are active at multiple layers when aggregating over tokens is relatively large, between 54 and 88%, and broadly increases with the model size for fixed hyperparameters (Figure 5). This measure quantifies the observation that, in the aggregate heatmaps (Figure 2), the distributions of latent activations over layers become more 'spread out' as the model size increases. Conversely, we find that the fraction of the variance for an individual latent explained by individual tokens is relatively small, on the order of 1 to 10%. This quantifies the observation that, in the single-prompt heatmaps (Figure 3), the distributions over layers are much less 'spread out' than in the aggregate heatmaps.

## 4.4 TUNED LENS

Thus far, we have assumed that the residual stream basis is the same at every layer. We relaxed this assumption by applying pre-trained tuned-lens transformations to the residual stream activations at each layer before the encoder (Section 3.3). We had expected that these transformations would increase the degree to which latents were active at multiple layers because they translate the activations at every layer into a basis more similar to the basis of the output layer. The aggregate and single-prompt heatmaps (Figures 6 and 7) indicate a modest increase in the degree to which latents are active at multiple layers compared with the standard approach.

The variance ratios in Figure 5 clarify that the tuned-lens approach decreases the degree to which latents are active at multiple layers when aggregating over tokens. This ratio remains approximately constant as the expansion factor increases, between 37% and 41% (Figure 27). Conversely, the variances for a single token relative to a single latent are larger, i.e., the single-prompt heatmaps are more 'spread out' compared with the standard approach.

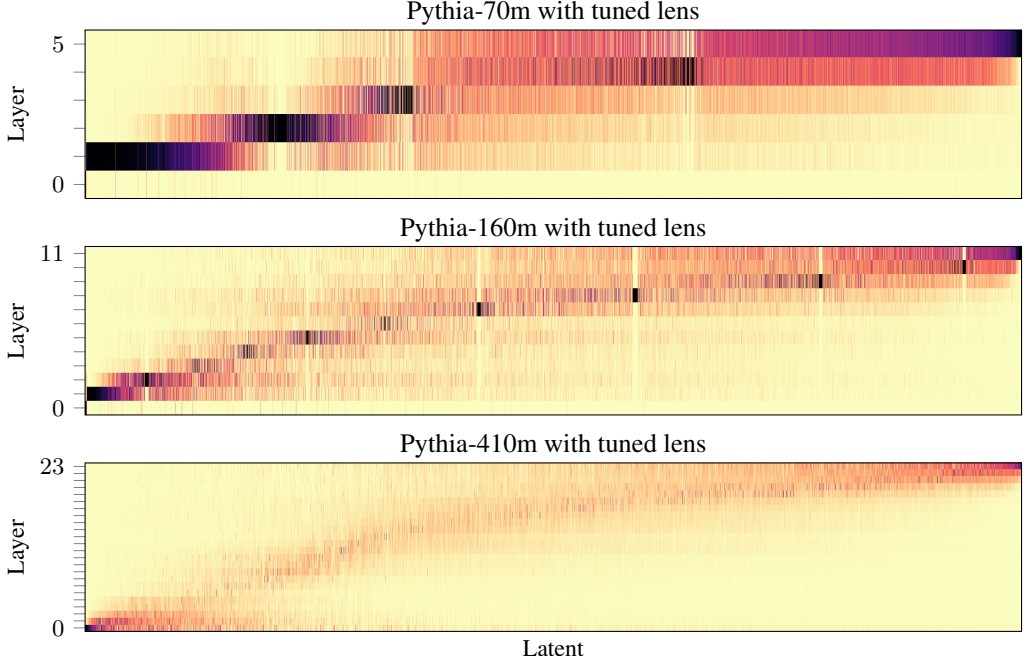

Figure 6: Heatmaps of the distributions of latent activations over layers when aggregating over 10 million tokens from the test set. Here, we plot the distributions for tuned-lens MLSAEs trained on Pythia models with an expansion factor of $R = 64$ and sparsity $k = 32$. For standard MLSAEs, see Figure 2. We note that a pre-trained tuned lens was not available for Pythia-1b (Section 3.3).

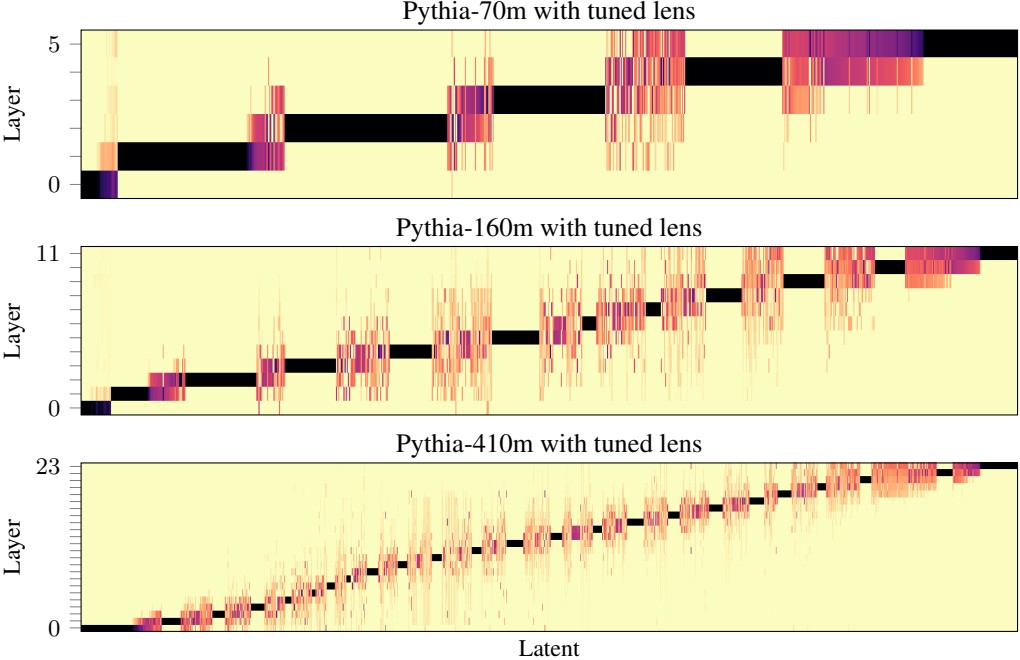

Figure 7: Heatmaps of the distributions of latent activations over layers for a single example prompt. Here, we plot the distributions for tuned-lens MLSAEs trained on Pythia models with an expansion factor of $R = 64$ and sparsity $k = 32$. The example prompt is "When John and Mary went to the store, John gave" (Wang et al., 2022). For standard MLSAEs, see Figure 3. We note that a pre-trained tuned lens was not available for Pythia-1b (Section 3.3).

## 5 DISCUSSION

We considered the activation vectors from different layers as different training examples, so we passed $n_L n_T$ vectors of length $d$ to the autoencoder, where $n_T$ is the number of tokens, $n_L$ is the number of layers, and $d$ is the dimension of the residual stream. This approach might be called a 'data-stacked' MLSAE. An alternative would be a 'feature-stacked' MLSAE, i.e., to concatenate the activation vectors from different layers into a single vector of dimension $n_L d$. This alternative might be better suited to capturing the notion of 'cross-layer superposition,' which we take to mean a small number of simultaneously active sparse features at multiple layers encoding a single meaningful concept (Olah, 2024; Templeton et al., 2024).

We began by pursuing the feature-stacked approach but discarded it. The essential issue is that a single set of sparse features describes the residual stream activations at every layer, which makes it difficult to understand how information flows through a transformer. For example, it would not be possible to plot the activations of sparse features across layers. Moreover, to compute this set of features, one must first compute the activations at every layer, which makes it more difficult to evaluate performance by traditional measures like single-layer reconstruction errors. Finally, the information encoded at one token position may differ substantially between layers due to self-attention. In the early layers, the representation is likely to primarily encode the input token and position embedding, whereas in the later layers, the representation may encode more complex properties of the surrounding context. It is not immediately apparent that jointly encoding this information by a single SAE is sensible. Instead, one might wish to separately capture the different information present at a token position across layers, which is allowed with our data-stacked approach.

## 6 CONCLUSION

We introduced the multi-layer SAE (MLSAE), where we train a single SAE on the activations at every layer of the residual stream. This allowed us to study both how information is represented within a single transformer layer and how information flows through the residual stream.

We confirmed that residual stream activations are relatively similar across layers by looking at cosine similarities before considering the distributions of latent activations over layers. When aggregating over a large sample of ten million tokens, we observed that most latents were active at multiple layers, but for a single prompt, most latent activations were isolated to a single layer. To quantify these observations, we computed the fraction of the total variance explained by individual latents and the fraction of the variance for an individual latent explained by individual tokens. This analysis confirmed that the degree to which latents are active at multiple layers when aggregating over tokens was large, increasing with the model size and expansion factor, and that the fraction of the variance explained by individual tokens was small.

Understanding how representations change as they flow through transformers is critical to identifying meaningful circuits, which is a core task of mechanistic interpretability. Despite the utility of the residual stream perspective, our results demonstrate that representation drift, and perhaps the increasing magnitude of changes to the residual stream across layers, is a significant obstacle to identifying meaningful computational variables with SAEs. Nevertheless, we argue that an approach such as the MLSAE, which considers the representations at multiple layers in parallel, is necessary for future methods that seek to interpret the internal computations of transformer language models.

## 7 ACKNOWLEDGEMENTS

Tim Lawson and Lucy Farnik were supported by the UKRI Centre for Doctoral Training in Interactive Artificial Intelligence (EP/S022937/1). This work was carried out with HPC systems provided by the Advanced Computing Research Centre at the University of Bristol. We also thank Dr. Stewart, whose philanthropy supported the compute resources used.

## 8 REPRODUCIBILITY STATEMENT

We release our code to train and analyze MLSAEs at `https://github.com/tim-lawson/mlsae`, and the models described in the paper at `https://huggingface.co/papers/2409.04185`. Section 3 and Appendix A describe the training setup, Section 4.1 and Appendix B describe the evaluation metrics, and Section 4.3 and Appendix C describe our analyses.

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

# A  TRAINING

We train each autoencoder on 1 billion tokens from the Pile (Gao et al., 2020), excluding the copyrighted Books3 dataset,[3] for a single epoch. Specifically, we concatenate a batch of 1024 text samples with the end-of-sentence token, tokenize the concatenated text, and divide the output into sequences of 2048 tokens, discarding the final incomplete sequence. We use an effective batch size of 131072 tokens (64 sequences) for all experiments.

We do not compute activation vectors and cache them to disk before training, which minimizes storage overhead at the expense of repeated computation. We construct a batch of activation vectors to input to the autoencoder by performing the forward pass of the underlying transformer for a sequence of tokens, collecting the residual stream activation vectors at every layer, and stacking them together. Following Lieberum et al. (2024), we exclude activation vectors corresponding to special tokens (end-of-sentence, beginning-of-sentence, and padding). Hence, each batch has an equal number of activation vectors from each layer, which is the number of non-special tokens.

Following the optimization guidelines in Bricken et al. (2023); Gao et al. (2024), we initialize the pre-encoder bias $\mathbf{b}_{\text{pre}}$ to the geometric median of the first training batch; we initialize the decoder weight matrix $\mathbf{W}_{\text{dec}}$ to the transpose of the encoder $\mathbf{W}_{\text{enc}}$; we scale the decoder weight vectors to unit norm at initialization and after each training step; and we remove the component of the gradient of the decoder weight matrix parallel to its weight vectors after each training step.

We use the Adam optimizer (Kingma & Ba, 2017) with the default $\beta$ parameters, a constant learning rate of $1 \times 10^{-4}$, and $\epsilon = 6.25 \times 10^{-10}$. Unlike Gao et al. (2024), we do not use gradient clipping or weight averaging, and we use FP16 mixed precision to reduce memory use.

# B  EVALUATION

## B.1  RECONSTRUCTION ERROR AND SPARSITY

While we use the FVU instead of MSE as the reconstruction error in the training loss, we record both metrics for the inputs from each transformer layer and the mean over all layers (Figure 8). The $L^0$ norm of the latents is fixed at $k$ per activation vector (layer and token), but we record the $L^1$ norm (Figure 9). We report the values of these metrics over one million tokens from the test set.

For Pythia-70m, the FVU at each layer is comparable to Marks et al. (2024, p. 21), who trained separate SAEs with $n = 32768$ and $L^0$ norms between 54 and 108, as well as Cunningham et al. (2023, p. 13). For Pythia-160m, the FVU is similar to Gao et al. (2024), who report the normalized MSE on layer 8 of GPT-2 small.

## B.2  DOWNSTREAM LOSS

In addition to the increase in cross-entropy (CE) loss, we record the Kullback-Leibler (KL) divergence between probability distributions when the residual stream activations at a given layer are replaced by their reconstruction (Section 4.1). We report the values of these metrics over one million tokens from the test set (Figure 10). The increase in cross-entropy loss is comparable to Marks et al. (2024, p. 21) for Pythia-70m, Gao et al. (2024, p. 5) and Braun et al. (2024) for GPT-2 small, and Lieberum et al. (2024, p. 7-8) for layer 20 of Gemma 2 2B and 9B.

## B.3  OTHER TRANSFORMER ARCHITECTURES

We predominantly study transformers from the Pythia suite because it affords a controlled setting to study how our results scale across model sizes (Biderman et al., 2023). While we do not expect our results to depend strongly on the underlying transformer architecture, we validated this assumption by training MLSAEs on Gemma 2 2B (Riviere et al., 2024), Llama 3.2 3B (Grattafiori et al., 2024), and GPT-2 small (Radford et al., 2019) with our default hyperparameters, i.e., an expansion factor of $R = 64$ and sparsity $k = 32$. We note that Gemma 2 2B and Llama 3.2 3B are similar in size to Pythia-2.8b, and GPT-2 small is similar in size to Pythia-160m.

---

[3] https://huggingface.co/datasets/monology/pile-uncopyrighted

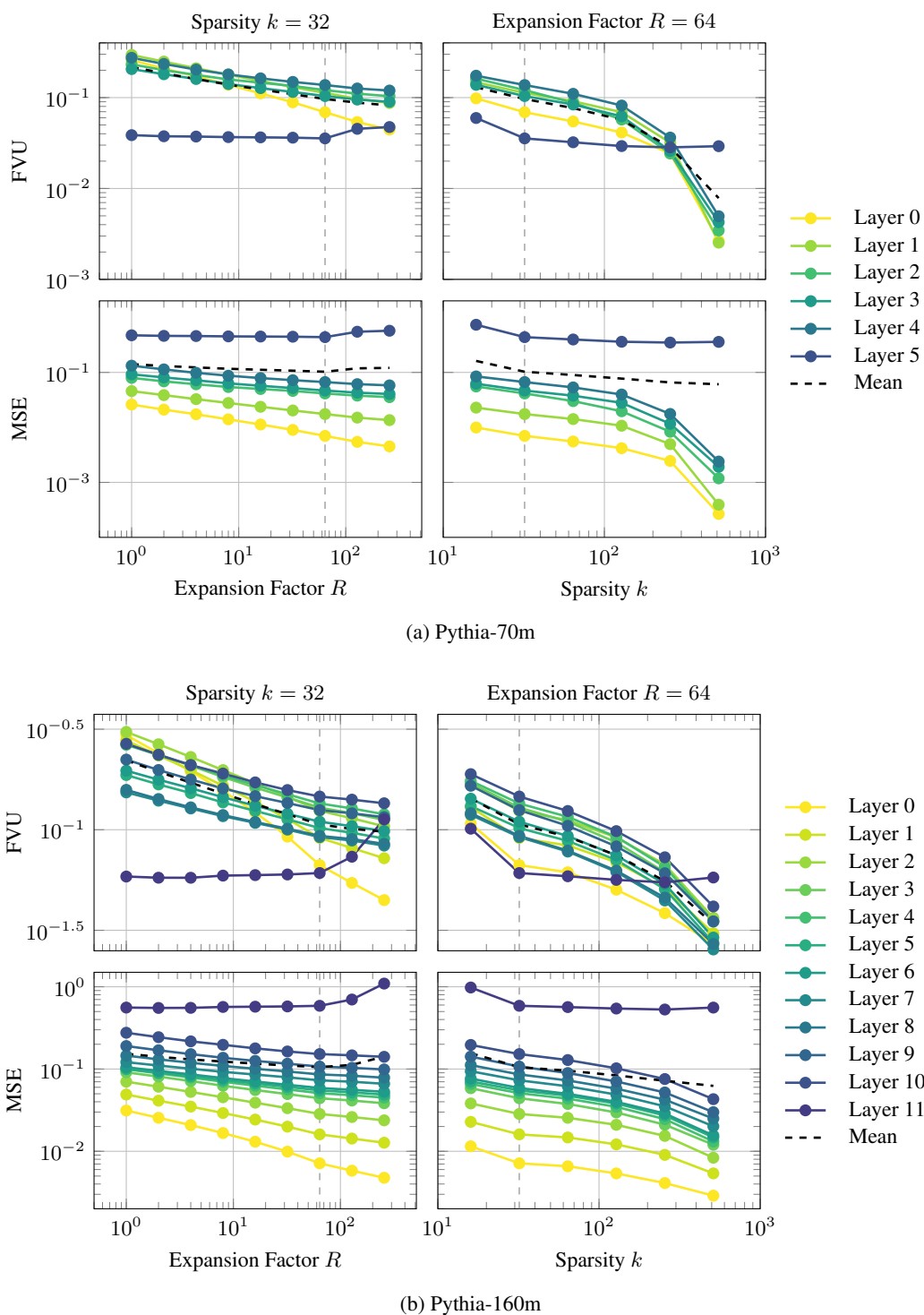

(a) Pythia-70m

(b) Pythia-160m

Figure 8: With fixed sparsity $k = 32$, the FVU and MSE generally decrease as the expansion factor $R$ increases. For inputs from the last layer, they increase for the largest expansion factors, which we attribute to fluctuations in the percentage of dead latents (Figure 11). With fixed expansion factor $R = 64$, the FVU and MSE decrease as the sparsity $k$ increases. While all inputs are standardized before passing them to the encoder, the decoder outputs are rescaled afterward. Hence, the MSE increases across layers because it is not divided by the variance of the inputs.

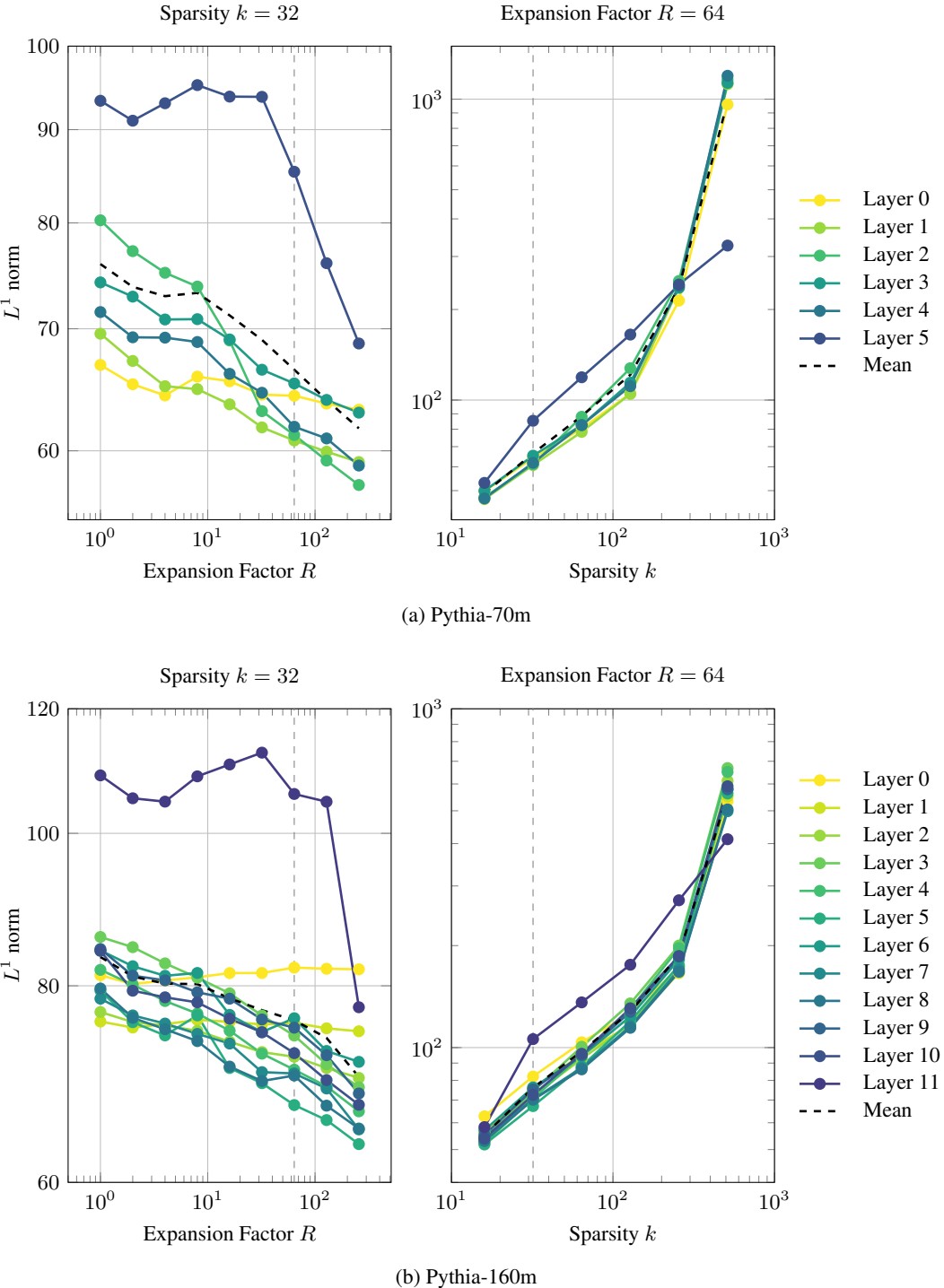

Figure 9: With fixed sparsity $k = 32$, the $L^1$ norm per token (the sum of absolute activations) generally decreases as the expansion factor $R$ increases. With fixed expansion factor $R = 64$, the $L^1$ norm increases as the sparsity $k$ increases. Recall that the $L^0$ norm per token (the count of non-zero activations) is fixed at $k$.

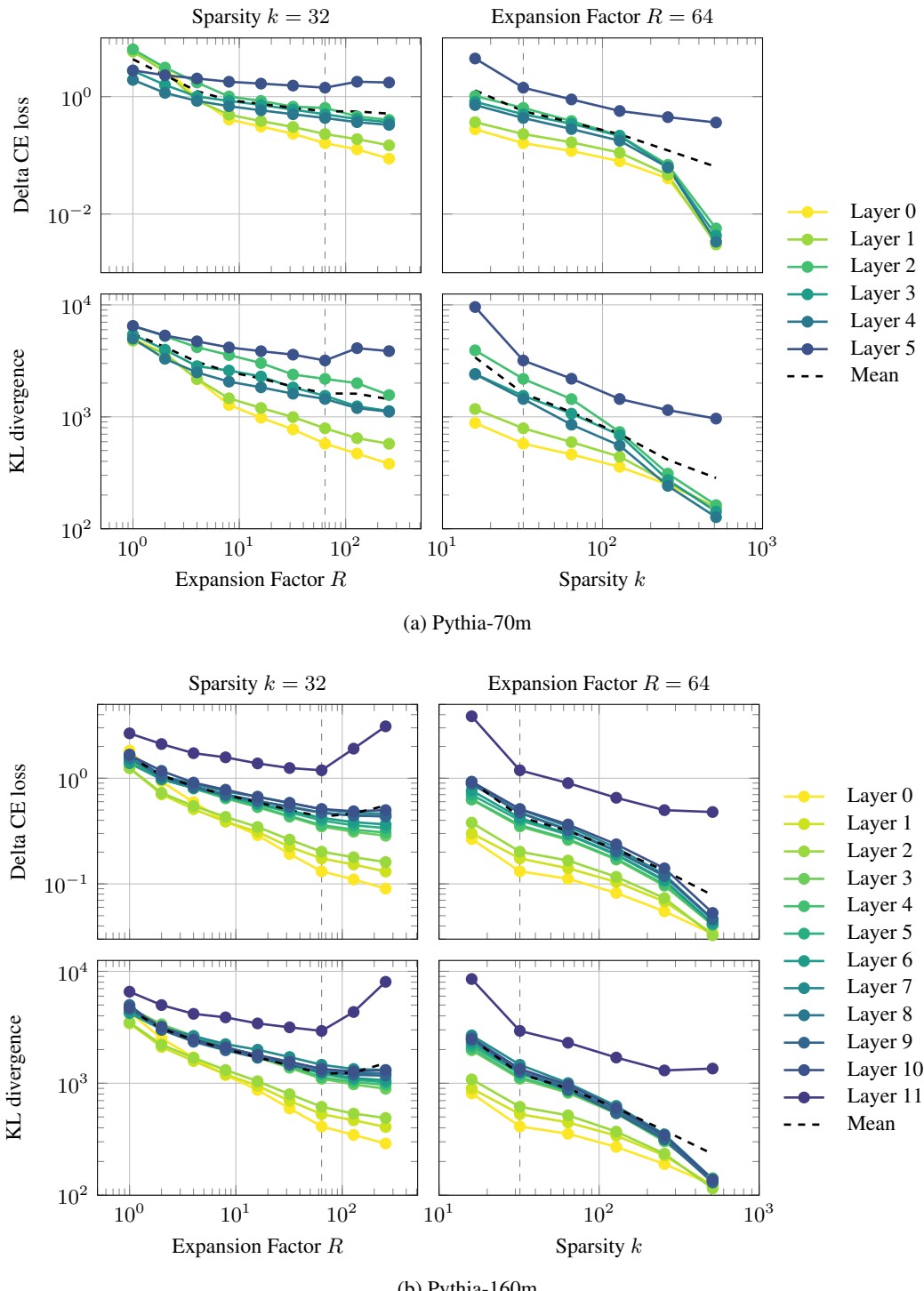

(a) Pythia-70m

(b) Pythia-160m

Figure 10: With fixed sparsity $k = 32$, the delta CE loss and KL divergence generally decrease as the expansion factor increases, except for inputs from the last layer. With fixed expansion factor $R = 64$, both metrics decrease as the sparsity $k$ increases, similarly to the FVU and MSE (Figure 8).

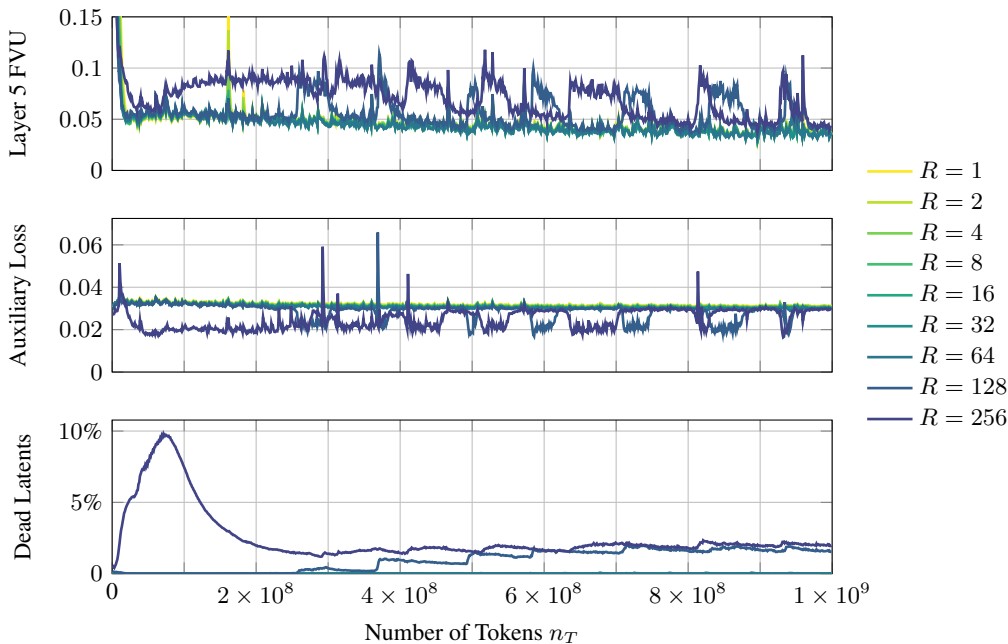

Figure 11: An illustration of the FVU for inputs from the last layer, compared to the auxiliary loss and percentage of dead latents, for MLSAEs trained on Pythia-70m with fixed sparsity $k = 32$. An increase in dead latents correlates with a decrease in the auxiliary loss and an increase in the FVU at the last layer. We attribute this to the increased scale of the inputs because the auxiliary loss depends on the MSE (Figure 8). The auxiliary loss is multiplied by its coefficient $\alpha = 1/32$ in the training loss.

We include quantitative results for Gemma 2 2B, Llama 3.2 3B, and GPT-2 small in Table 1, Figure 5, and throughout Appendix B. We include heatmaps of the distributions of latent activations over layers in Figures 13, 14, 15, 16, 17, and 18, which are qualitatively similar to the Pythia models. Interestingly, we observed that our validation metrics improved at a slower rate for Gemma 2 2B, so we continued to train this MLSAE up to a total of approximately 2.5 billion tokens. Nevertheless, the increase in the cross-entropy loss remains larger than other models of similar sizes (Table 1).

## B.4 SINGLE-LAYER SAEs

While we compare the performance of our multi-layer SAEs to single-layer SAEs from the literature in Appendix B.1 and B.2, we also trained multiple single-layer SAEs on Pythia-70m, 160m, and 410m with our default hyperparameters, leaving the remainder of the experimental setup unchanged.

Predictably, we find that a single-layer SAE trained on data from a given layer performs best on test data from the same layer (Figures 19, 20, and 21). A multi-layer SAE trained on data from every layer performs comparably to the corresponding single-layer SAE, and more consistently across test data from different layers. Interestingly, applying the corresponding tuned-lens transformation to the input activations from each layer during training and evaluation degrades the performance of single-layer SAEs on test data from different layers of Pythia-70m, unlike multi-layer SAEs (Figure 12).

Importantly, the results for the last layer are excluded from these figures. This is because we take the residual stream activation vectors after a given layer has been applied (Section 3.1), such that the last-layer activations represent only the next-token predictions of the model and not intermediate computational variables. Hence, we expect these activations to have a significantly different structure to the preceding layers, which could distort comparisons between layers (Lad et al., 2024).

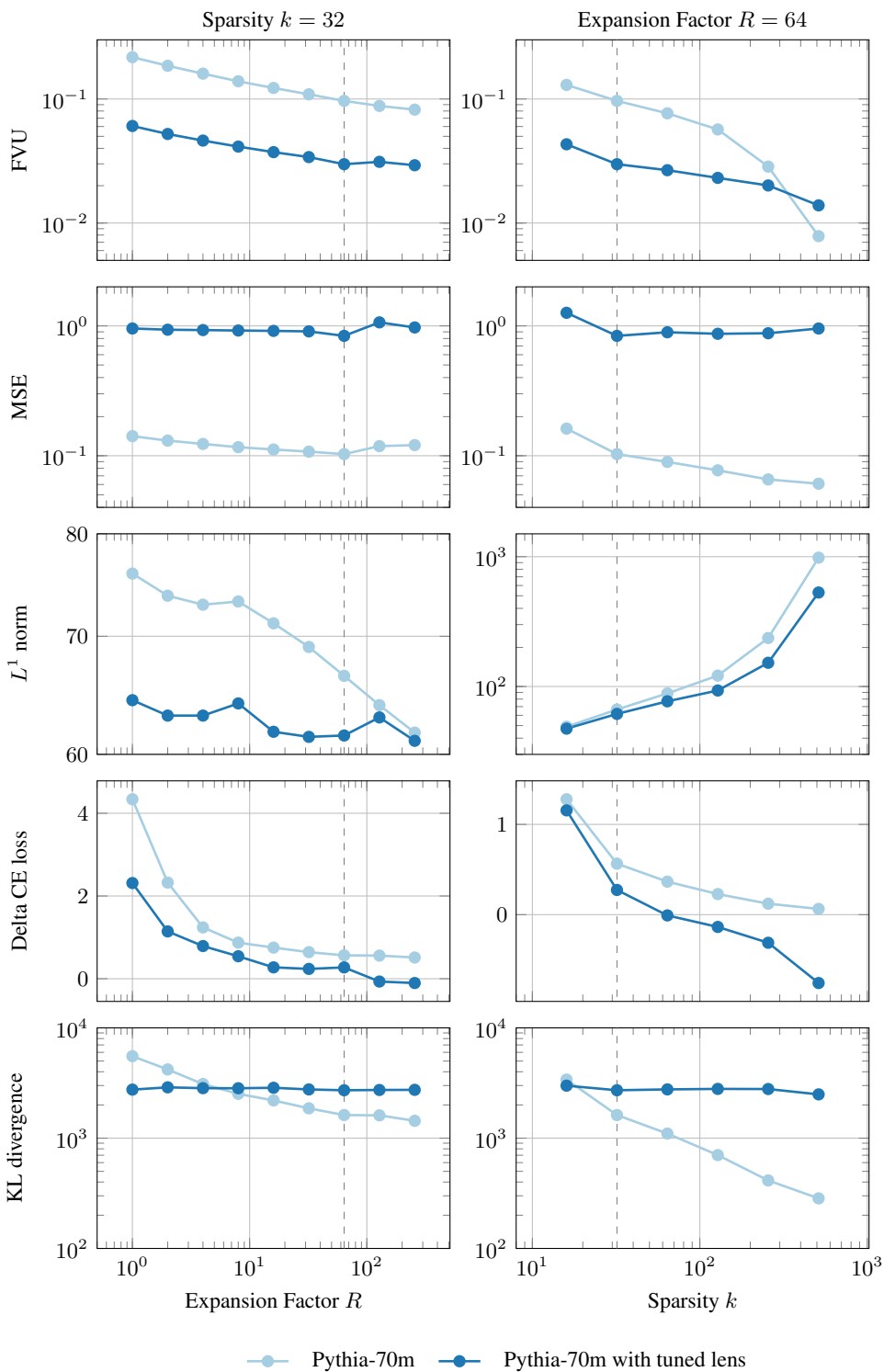

Figure 12: For Pythia-70m, applying tuned-lens transformations decreases the mean FVU and delta cross-entropy loss but not the KL divergence. Importantly, we compute reconstruction errors before applying the inverse transformation and downstream loss metrics afterward (Section 3.3). Unlike Figure 10, we use a linear scale for the delta cross-entropy loss because, surprisingly, it is negative for tuned-lens MLSAEs with a large expansion factor $R$ or sparsity $k$; see also Figure 22.

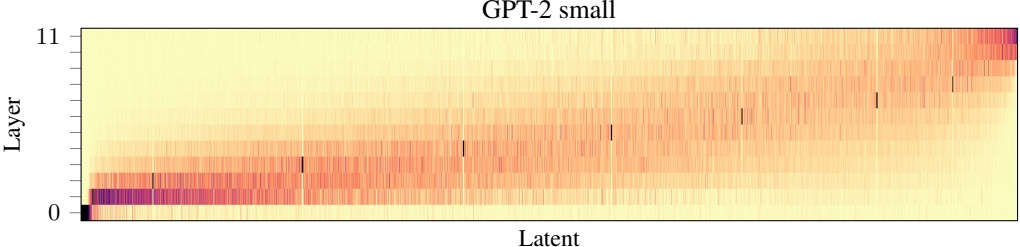

Figure 13: Heatmaps of the distributions of latent activations over layers when aggregating over 10 million tokens from the test set. Here, we plot the distributions for MLSAEs trained on GPT-2 small with an expansion factor of $R = 64$. We provide further details in Figure 2.

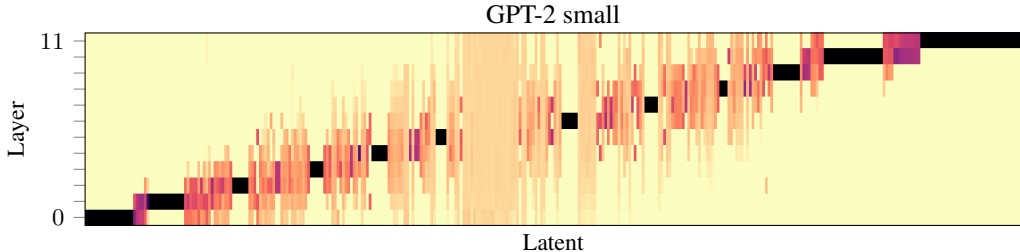

Figure 14: Heatmaps of the distributions of latent activations over layers for a single example prompt. Here, we plot the distributions for MLSAEs trained on GPT-2 small with an expansion factor of $R = 64$. The example prompt is "When John and Mary went to the store, John gave" (Wang et al., 2022). We provide further details in Figure 3.

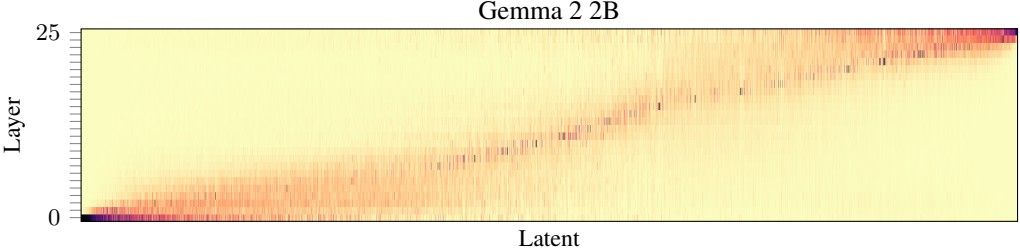

Figure 15: Heatmaps of the distributions of latent activations over layers when aggregating over 10 million tokens from the test set. Here, we plot the distributions for MLSAEs trained on Gemma 2 2B with an expansion factor of $R = 64$. We provide further details in Figure 2.

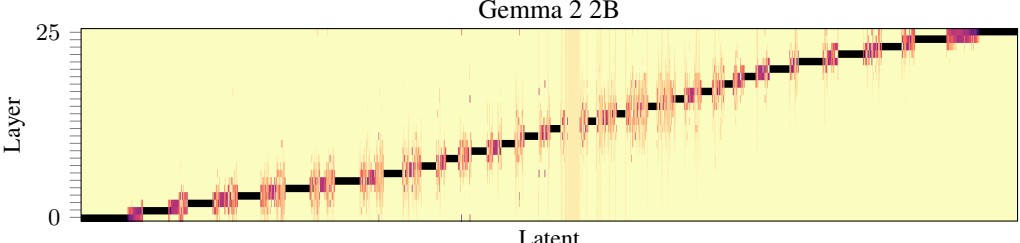

Figure 16: Heatmaps of the distributions of latent activations over layers for a single example prompt. Here, we plot the distributions for MLSAEs trained on Gemma 2 2B with an expansion factor of $R = 64$. The example prompt is "When John and Mary went to the store, John gave" (Wang et al., 2022). We provide further details in Figure 3.

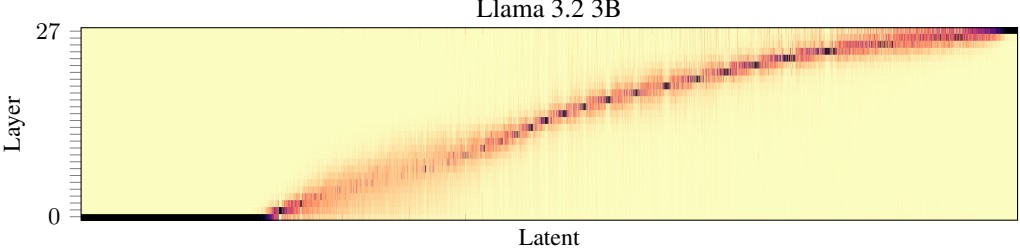

Figure 17: Heatmaps of the distributions of latent activations over layers when aggregating over 10 million tokens from the test set. Here, we plot the distributions for MLSAEs trained on Llama 3.2 3B with an expansion factor of $R = 64$. We provide further details in Figure 2. Notably, a greater proportion of latents are only active at the first layer compared with other transformer architectures.

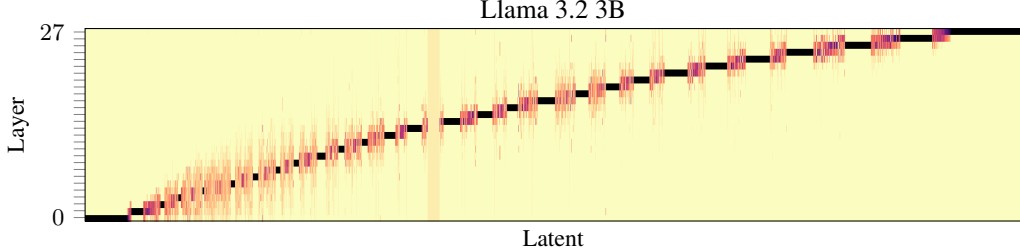

Figure 18: Heatmaps of the distributions of latent activations over layers for a single example prompt. Here, we plot the distributions for MLSAEs trained on Llama 3.2 3B with an expansion factor of $R = 64$. The example prompt is "When John and Mary went to the store, John gave" (Wang et al., 2022). We provide further details in Figure 3.

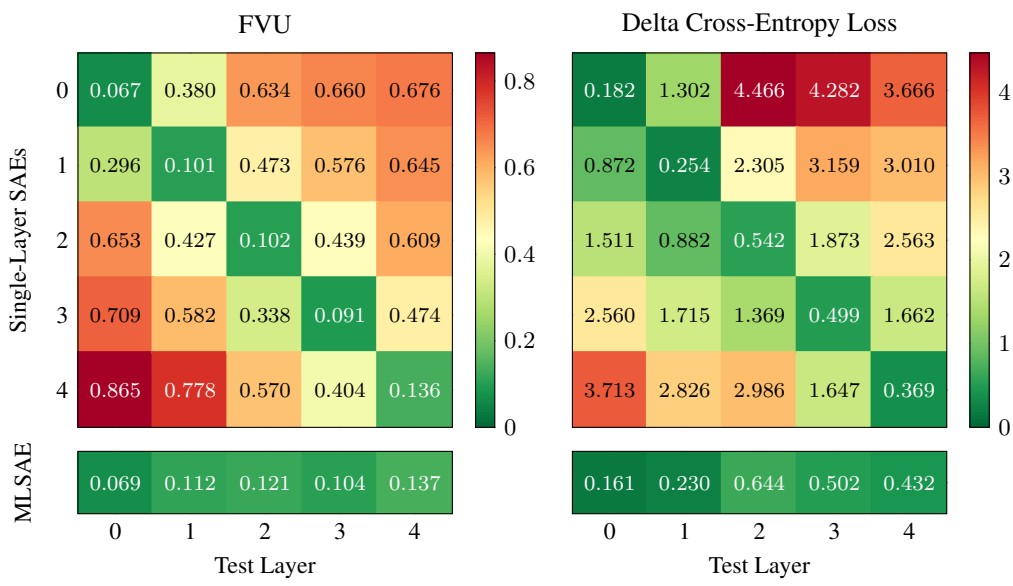

Figure 19: The FVU reconstruction error and delta cross-entropy loss for single-layer SAEs trained on each layer of Pythia-70m, compared with a single multi-layer SAE trained on every layer. The single-layer SAEs trained on data from a given layer perform best on test data from the same layer (the diagonal elements of the matrix plot); a multi-layer SAE trained on data from every layer performs comparably to the corresponding single-layer SAEs (the row beneath the matrix plot).

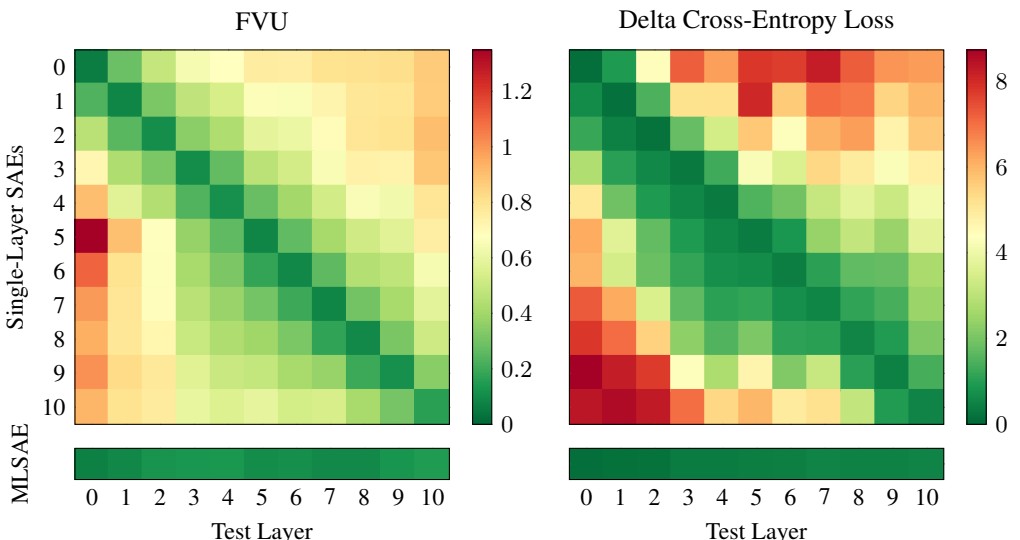

Figure 20: The FVU reconstruction error and delta cross-entropy loss for single-layer SAEs trained on each layer of Pythia-160m, compared with a single multi-layer SAE trained on every layer. The single-layer SAEs trained on data from a given layer perform best on test data from the same layer (the diagonal elements of the matrix plot); a multi-layer SAE trained on data from every layer performs comparably to the corresponding single-layer SAEs (the row beneath the matrix plot).

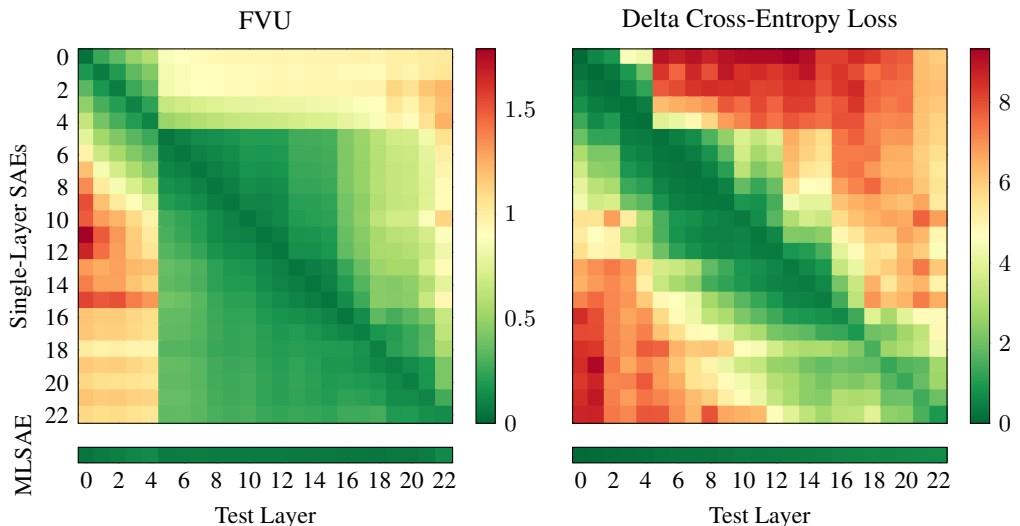

Figure 21: The FVU reconstruction error and delta cross-entropy loss for single-layer SAEs trained on each layer of Pythia-410m, compared with a single multi-layer SAE trained on every layer. The single-layer SAEs trained on data from a given layer perform best on test data from the same layer (the diagonal elements of the matrix plot); a multi-layer SAE trained on data from every layer performs comparably to the corresponding single-layer SAEs (the row beneath the matrix plot). Interestingly, there is a sharp change in the FVU between single-layer SAEs trained on layers 4 and 5.

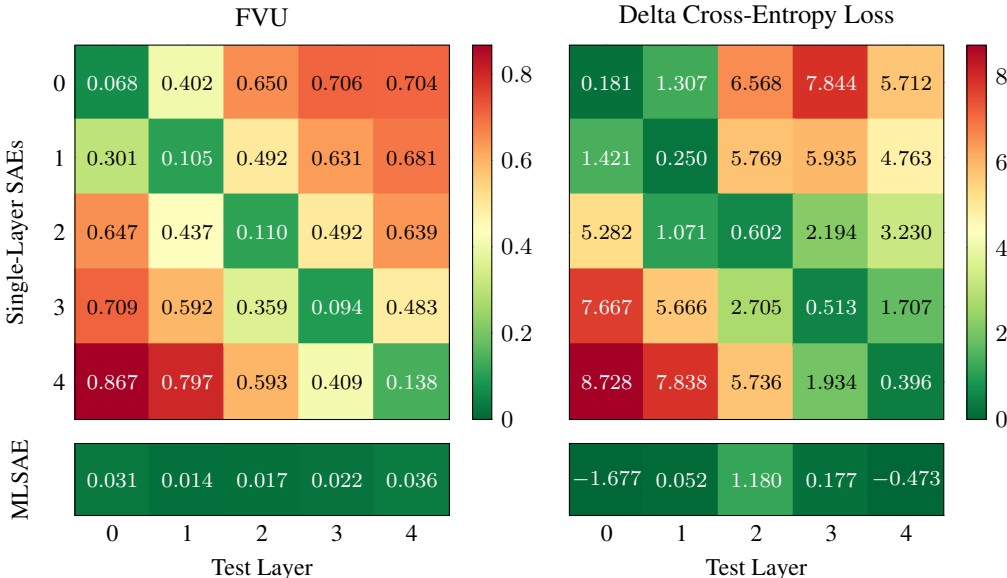

Figure 22: The FVU reconstruction error and delta cross-entropy loss for single-layer SAEs trained on each layer of Pythia-70m compared with a single multi-layer SAE trained on every layer, applying tuned-lens transformations during training and evaluation (Section 3.3).

## B.5 MEAN MAX COSINE SIMILARITY

Sharkey et al. (2022) define the Mean Max Cosine Similarity (MMCS) between a learned dictionary $X$ and a ground-truth dictionary $X'$. There is no ground-truth dictionary for language models, so a larger learned dictionary or the $k$ nearest neighbors to each dictionary element are commonly used.

$$\text{MMCS}(X, X') = \frac{1}{|X|} \sum_{\mathbf{x} \in X} \max_{\mathbf{x}' \in X'} \cos \text{sim}(\mathbf{x}, \mathbf{x}') \tag{13}$$

The MMCS serves as a proxy measure for 'feature splitting' (Bricken et al., 2023; Braun et al., 2024): as the number of features increases, we expect the decoder weight vectors to be more similar to their nearest neighbors. We compute the MMCS with $k = 1$ after training, finding it decreases slightly as the model size increases with fixed hyperparameters (Table 2).

## B.6 PAIRWISE COSINE SIMILARITIES

A potential issue when training multi-layer SAEs is learning multiple versions of 'the same' latent, i.e., multiple latents with similar interpretable functions but which are active at different layers. In this case, we would expect to find pairs of latents with relatively large cosine similarities between their decoder weight vectors but different observed distributions of activations over layers (Section 4.3). We investigated this possibility by comparing the distributions of pairwise cosine similarities between decoder weight vectors for trained MLSAEs to reference distributions.

As a negative control, we generated an equal number (the number of latents $n$) of normal independently and identically distributed (i.i.d.) vectors $\mathbf{x} \sim \mathcal{N}(\mathbf{0}, \mathbf{I})$ of the same length (the model dimension $d$). In this case, the pairwise cosine similarities follow a normal distribution $\cos \text{sim}(\mathbf{x}, \mathbf{x}') \sim \mathcal{N}(0, 1/d)$. As a positive control, we generated a smaller number of normal i.i.d. vectors (the number of latents $n$ divided by the number of layers $n_L$), copied the vectors $n_L$ times, and added noise $\sim \mathcal{N}(0, 1)$ to each copy. In this case, we expect an additional frequency peak for large positive cosine similarities.

Figure 24 shows that the distributions of pairwise cosine similarities for decoder weight vectors are slightly heavier-tailed and right-shifted compared with the negative control, i.e., a pair of MLSAE latents are slightly more likely to have high cosine similarity than a pair of i.i.d. normal vectors. However, the number of pairs with large positive cosine similarities is small compared to the positive control, which has a second peak around 0.5 (visible only with the logarithmic $y$-axis scale).

| Model | Mean | Std. Dev. |
|---|---|---|
| Pythia-70m | 0.275 | 0.0843 |
| Pythia-160m | 0.250 | 0.0928 |
| Pythia-410m | 0.221 | 0.0868 |
| Pythia-1b | 0.201 | 0.0989 |
| Pythia-1.4b | 0.180 | 0.0861 |
| Gemma 2 2B | 0.249 | 0.1052 |
| Llama 3.2 3B | 0.215 | 0.1187 |
| GPT-2 small | 0.258 | 0.0703 |

(a) Standard

| Model | Mean | Std. Dev. |
|---|---|---|
| Pythia-70m | 0.261 | 0.0763 |
| Pythia-160m | 0.206 | 0.0734 |
| Pythia-410m | 0.216 | 0.0864 |

(b) Tuned lens

Table 2: The mean and standard deviation of the maximum cosine similarity between decoder weight vectors for MLSAEs with an expansion factor of $R = 64$ and sparsity $k = 32$. The MMCS decreases as the model size increases for Pythia transformers.

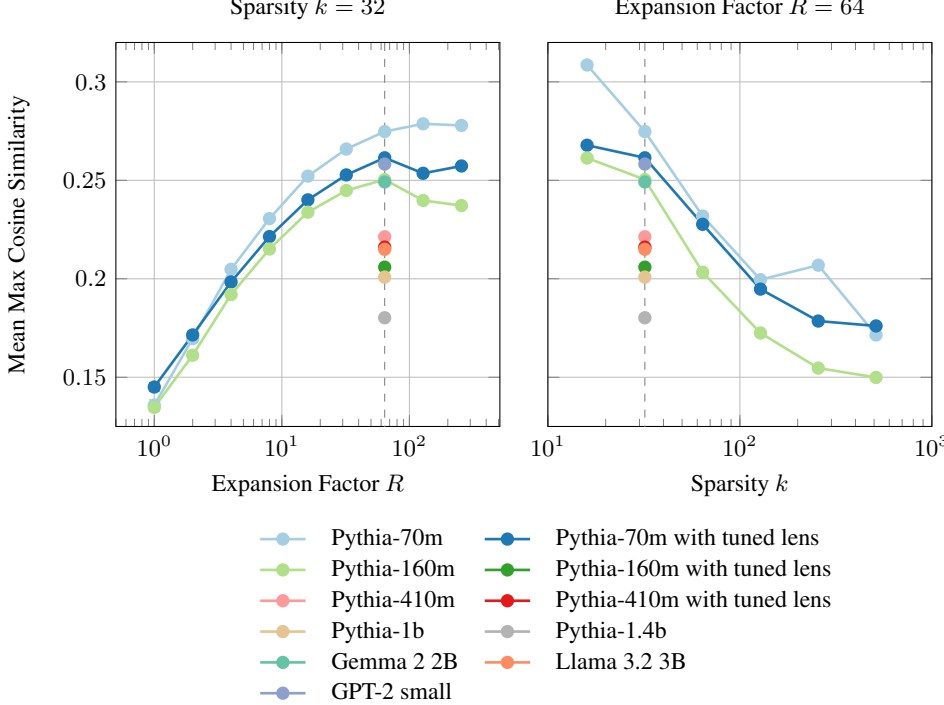

Figure 23: The Mean Max Cosine Similarity between decoder weight vectors for standard and tuned-lens MLSAEs. The MMCS increases as the expansion factor $R$ increases and decreases as the sparsity $k$ increases. Applying tuned-lens transformations slightly decreases the MMCS relative to standard MLSAEs.

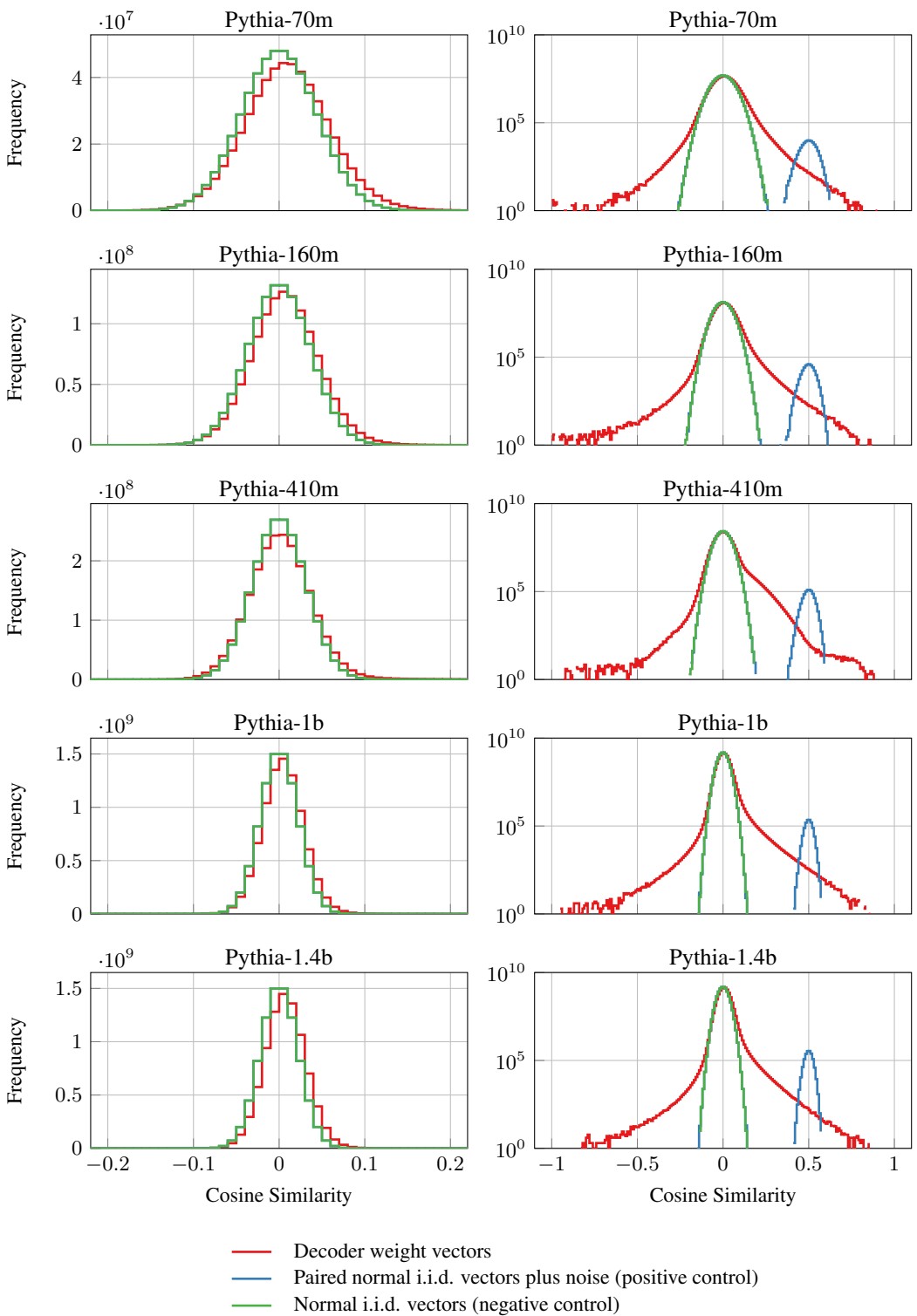

Figure 24: Histograms of the frequencies of pairwise cosine similarities between decoder weight vectors, compared to an equal number of normal i.i.d. vectors of the same length, and $n_L$ copies of a smaller number of normal i.i.d. vectors with added noise $\sim \mathcal{N}(0, 1)$. Here, we report the frequencies for MLSAEs trained on Pythia models with an expansion factor of $R = 64$ and sparsity $k = 32$. The left-hand $y$-axis scale is linear, the right-hand is logarithmic.

## C  MEASURES OF LATENTS ACTIVE AT MULTIPLE LAYERS

### C.1  HEATMAP NORMALIZATION

In the aggregate and single-prompt heatmaps such as Figures 2 and 3, we plot the distributions of latent activations over layers, which we take to be proportional to the total activations when aggregating over a large sample of tokens (Eq. 10). We normalized the latent activations in this way to visually compare the aggregate and single-prompt heatmaps, as well as individual latents within a heatmap, due to the wide range of activation counts and totals across latents.

Normalizing the activations discards the relative frequencies and magnitudes of activations for different latents, so we reproduce Figures 2 and 3 with the un-normalized totals of latent activations in Figures 25 and 26. We use power-law normalization for the colormaps, i.e., $y = x^\gamma$, to account for the wide range of values; all other heatmaps have linear colormaps. As with all other single-prompt heatmaps, we exclude latents from Figure 26 that never activate. In both cases, the un-normalized results are qualitatively similar to the normalized results.

### C.2  VARIANCE OF THE LAYER INDEX

Recall that we consider the layer $L$, token $T$, and latent index $J$ as random variables (Section 4.3). For a single latent, we have, by the law of total variance:

$$\mathrm{Var}\,[L] = \mathbb{E}[\mathrm{Var}\,[L \mid T]] + \mathrm{Var}\,[\mathbb{E}[L \mid T]] \tag{14}$$

We are interested in the first two terms:

- $\mathrm{Var}\,[L]$ is the variance of the distribution over layers, aggregating over tokens;
- $\mathbb{E}[\mathrm{Var}\,[L \mid T]]$ is the mean variance of the distributions over layers for each token; and
- $\mathrm{Var}\,[\mathbb{E}[L \mid T]]$ is the variance of the mean layers for each token.

Aggregating over latents, we have:

$$\mathbb{E}[\mathrm{Var}\,[L \mid J]] = \mathbb{E}[\mathrm{Var}\,[L \mid T, J]] + \mathbb{E}[\mathrm{Var}\,[\mathbb{E}[L \mid T, J] \mid J]] \tag{15}$$

### C.3  NUMBER OF LAYERS ABOVE A THRESHOLD

The count of layers at which a latent is non-zero ('active') does not necessarily positively correlate with the variance of the layer index considered in Section 4.3. For example, the variance of 0 and 5 (two distinct values) is greater than the variance of 2, 3, and 4 (three distinct values). Strictly speaking, the layer index is ordinal data, but we implicitly treat it as interval data by taking the arithmetic mean and variance. We chose this approach because we expected latents to be active over a contiguous range of layers, which is validated qualitatively by the heatmaps such as Figures 2 and 3.

For comparison, we computed the number of layers at which each latent has a count of non-zero activations above a threshold ('active layers') divided by the total number of model layers $n_L$. We selected a threshold count of 10k tokens (0.1% of a sample of 10M tokens). When aggregating over latents, the relative mean active layers decreases as the model size increases for Pythia models (Figure 28a) and as the number of latents increases relative to the model dimension (Figure 28b). Importantly, this measure depends strongly on the choice of threshold, unlike our variance ratios.

### C.4  ENTROPY

A further measure of the degree to which a latent is active at multiple layers is the statistical distance between the observed discrete distribution of activations over layers (Eq. 10) and a reference distribution. At one extreme is a Dirac distribution with probability mass 1 for a single layer index and 0 elsewhere, in which case the latent is active at a single layer. The other extreme is the discrete uniform distribution $\mathcal{U}(0, n_L)$, in which case the latent is equally active at every layer. Hence, the entropy of the observed distribution ranges between 0 and $\ln n_L$. Notably, the entropy of the observed distribution is agnostic with respect to the numeric values of the layer indices and their ordering.

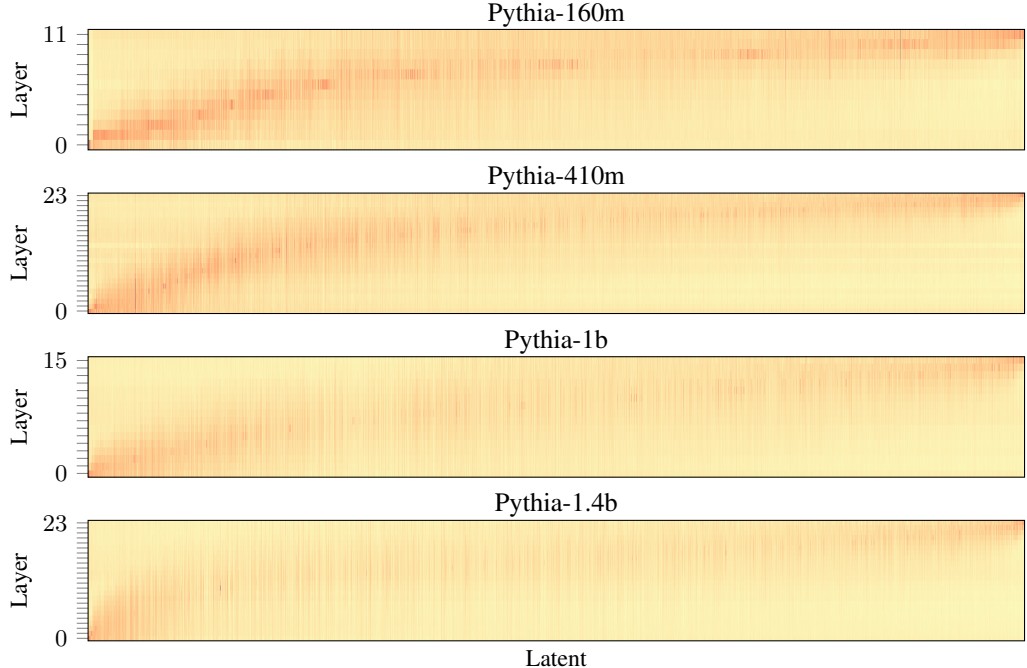

Figure 25: Heatmaps of the total latent activations over layers when aggregating over 10 million tokens from the test set. Here, we plot the totals for MLSAEs trained on Pythia models with an expansion factor of $R = 64$ and sparsity $k = 32$. We provide further details in Figure 2. The colormaps use power-law normalization with $\gamma = 1/4$.

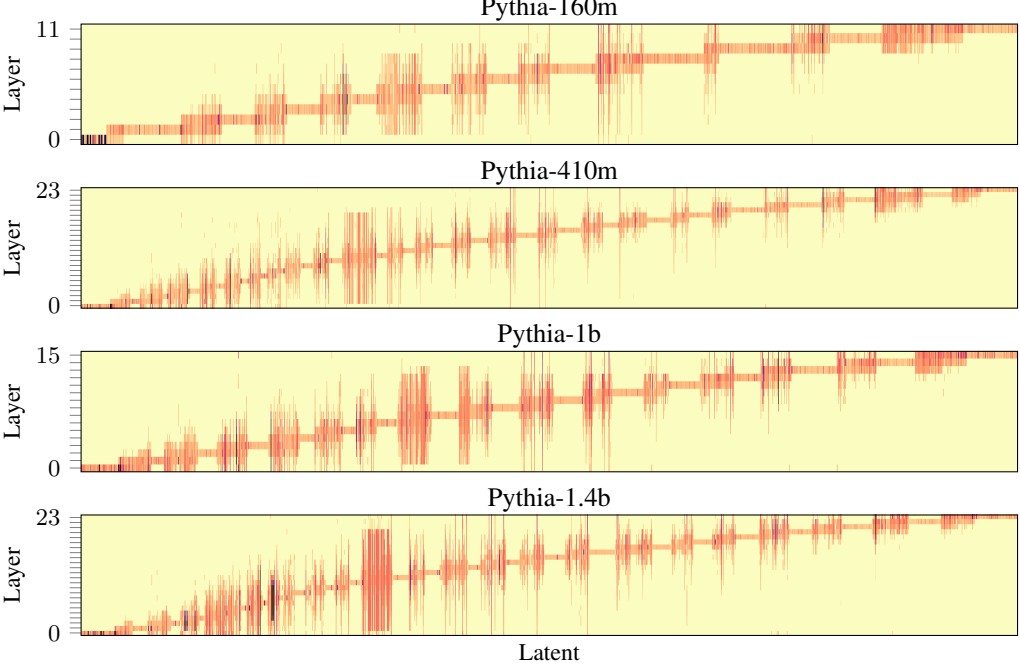

Figure 26: Heatmaps of the total latent activations over layers for a single example prompt. Here, we plot the totals for MLSAEs with an expansion factor of $R = 64$ and sparsity $k = 32$. The example prompt is "When John and Mary went to the store, John gave" (Wang et al., 2022). We provide further details in Figure 3. The colormaps use power-law normalization with $\gamma = 1/2$.

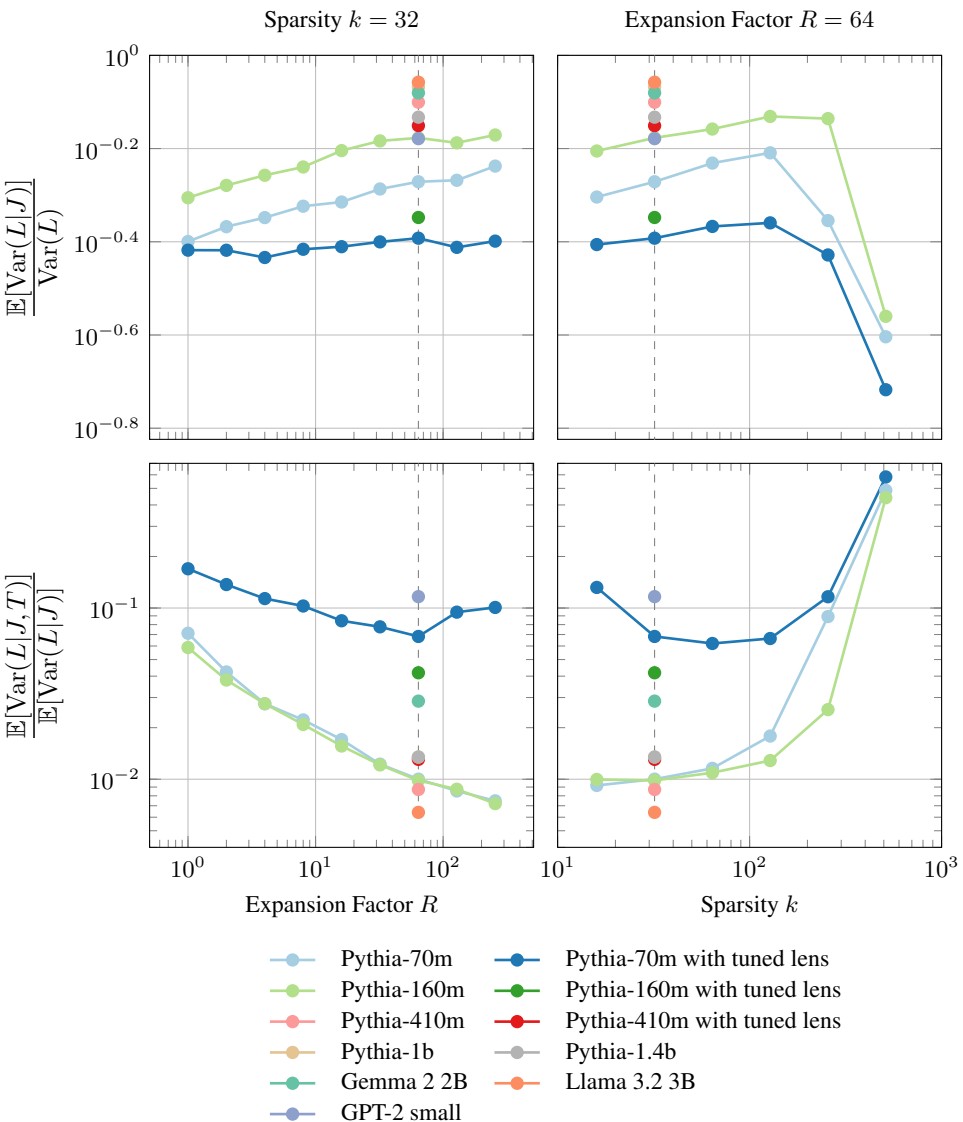

Figure 27: The fraction of the total variance explained by individual latents and the fraction of the variance for an individual latent explained by individual tokens (Eqs. 11 and 12). Here, we plot the variance ratios for standard and tuned-lens MLSAEs over 10 million tokens from the test set.

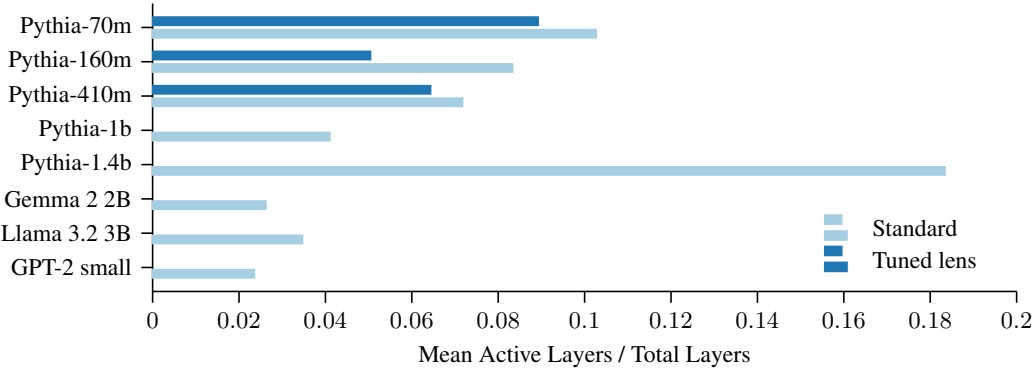

(a) Varying the model with an expansion factor of $R = 64$ and sparsity $k = 32$

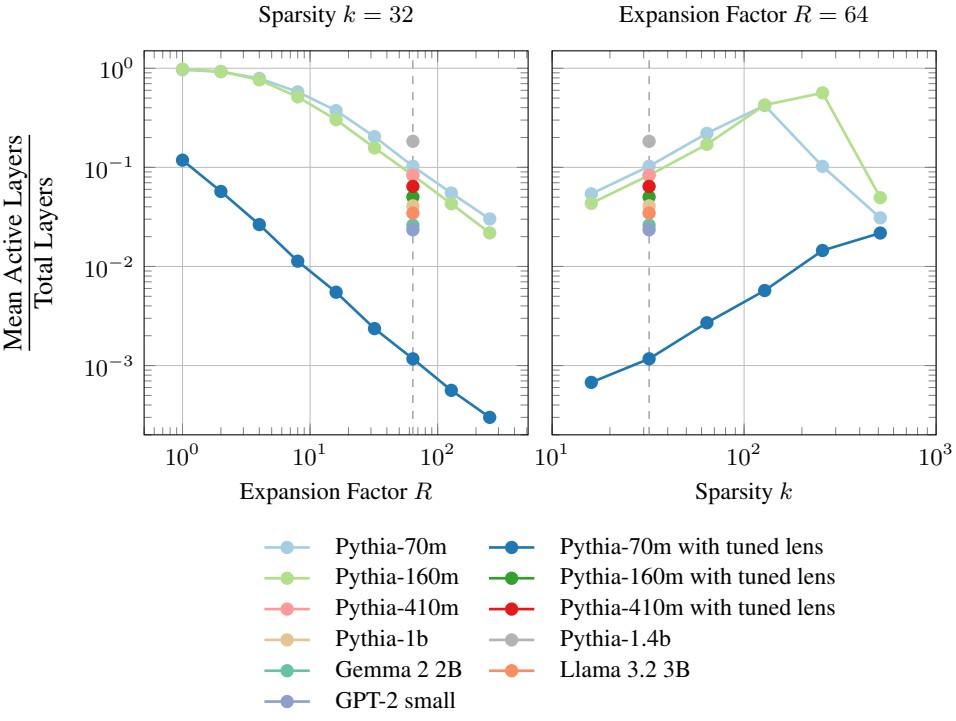

(b) Varying the expansion factor $R$ with sparsity $k = 32$ and $k$ with $R = 64$

Figure 28: The mean number of layers at which latents have a count of non-zero activations above a threshold divided by the total number of model layers, over 10 million tokens from the test set. The threshold is 10 thousand tokens (0.1%). As in Figure 5, the absence of bars for tuned-lens MLSAEs indicates the absence of results, not that the values are zero. The mean active layers decreases as the expansion factor $R$ increases, and increases as the sparsity $k$ increases up to a point. There is no clear trend with respect to the model size. Notably, applying tuned-lens transformations to the input activations from each layer decreases the mean active layers (Section 3.3).

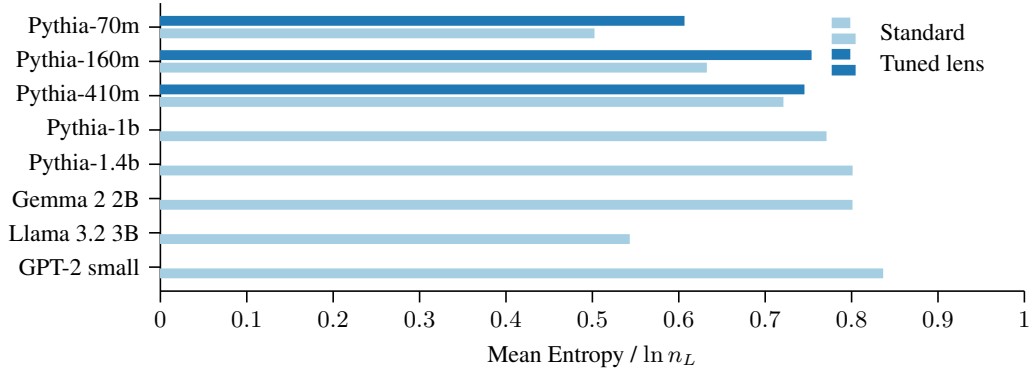

(a) Varying the model with an expansion factor of $R = 64$ and sparsity $k = 32$

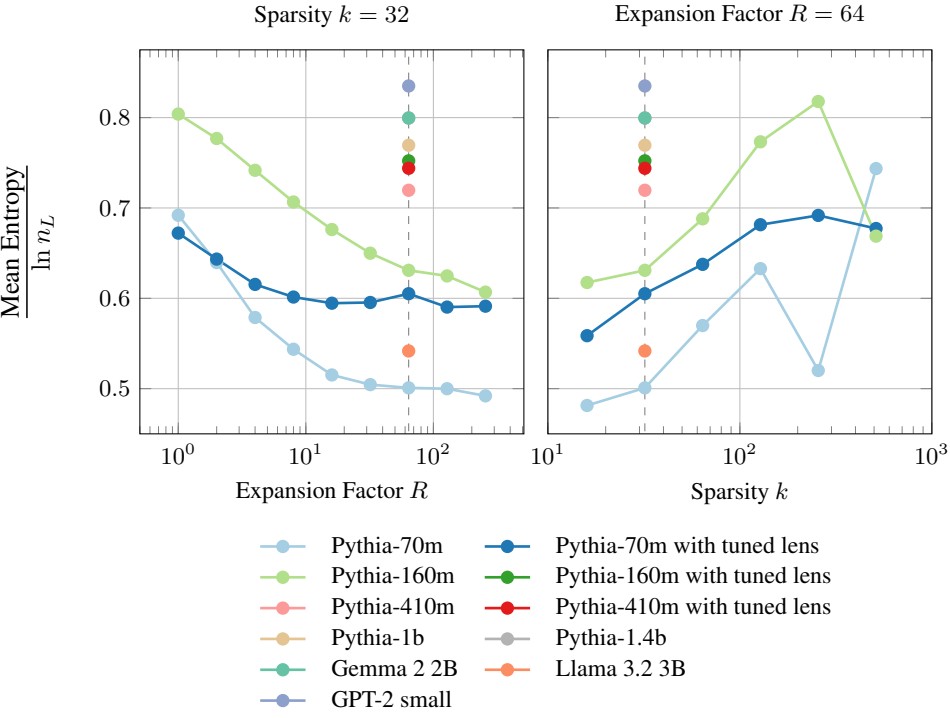

(b) Varying the expansion factor $R$ with sparsity $k = 32$ and $k$ with $R = 64$

Figure 29: The mean entropy of the observed discrete distributions of latent activations over layers (Eq. 10) divided by the maximum entropy $\ln n_L$, over 10 million tokens from the test set. As in Figure 5, the absence of bars for tuned-lens MLSAEs indicates the absence of results, not that the values are zero. The entropy tends to decrease as the expansion factor $R$ increases, to increase as the sparsity $k$ increases, and to increase as the model size increases for Pythia transformers.

We computed the entropy of the observed distributions of activations over layers, took the mean over latents, and divided it by $\ln n_L$ to compare models with different numbers of layers. The normalized mean entropy increases slightly as the model size increases for Pythia models (Figure 29a), similarly to the variance of the layer index (Section 4.3). However, it decreases as the number of latents increases relative to the model dimension, similarly to the mean active layers (Figure 29b).

## D    ADDITIONAL HEATMAPS

For completeness, we include equivalent aggregate and single-prompt heatmaps to Figures 2 and 3 for different models and combinations of hyperparameters:

- Varying $R$ for Pythia-70m and $k = 32$ (Figures 30 and 31)
- Varying $k$ for Pythia-70m and $R = 64$ (Figures 32 and 33)
- Varying $R$ for Pythia-160m and $k = 32$ (Figures 34 and 35)
- Varying $k$ for Pythia-160m and $R = 64$ (Figures 36 and 37)
- Varying $R$ for Pythia-70m with tuned lens and $k = 32$ (Figures 38 and 39)
- Varying $k$ for Pythia-70m with tuned lens and $R = 64$ (Figures 40 and 41)

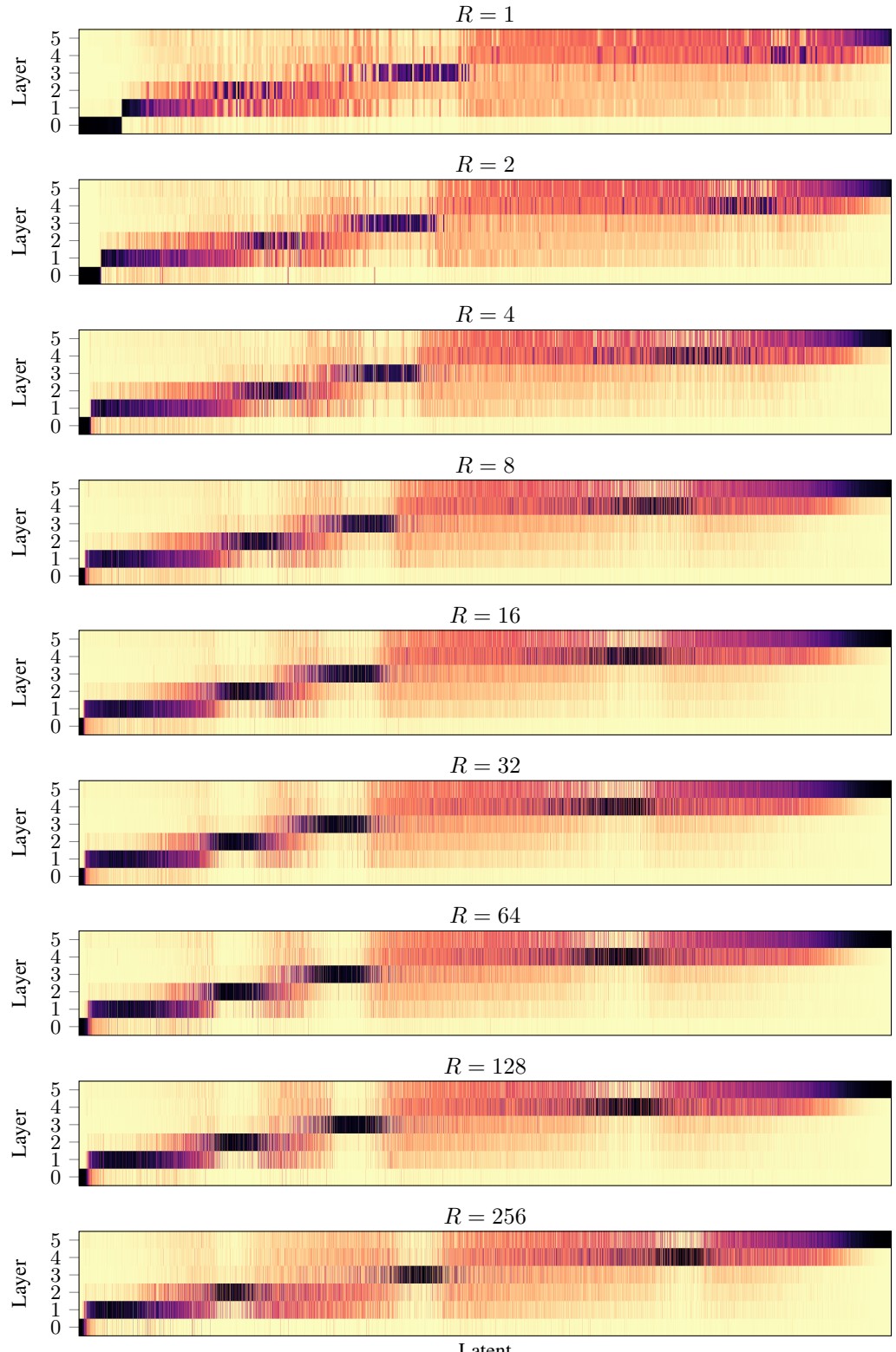

Figure 30: Heatmaps of the distributions of latent activations over layers when aggregating over 10 million tokens from the test set. Here, we plot the distributions for MLSAEs trained on Pythia-70m with sparsity $k = 32$. We provide further details in Figure 2.

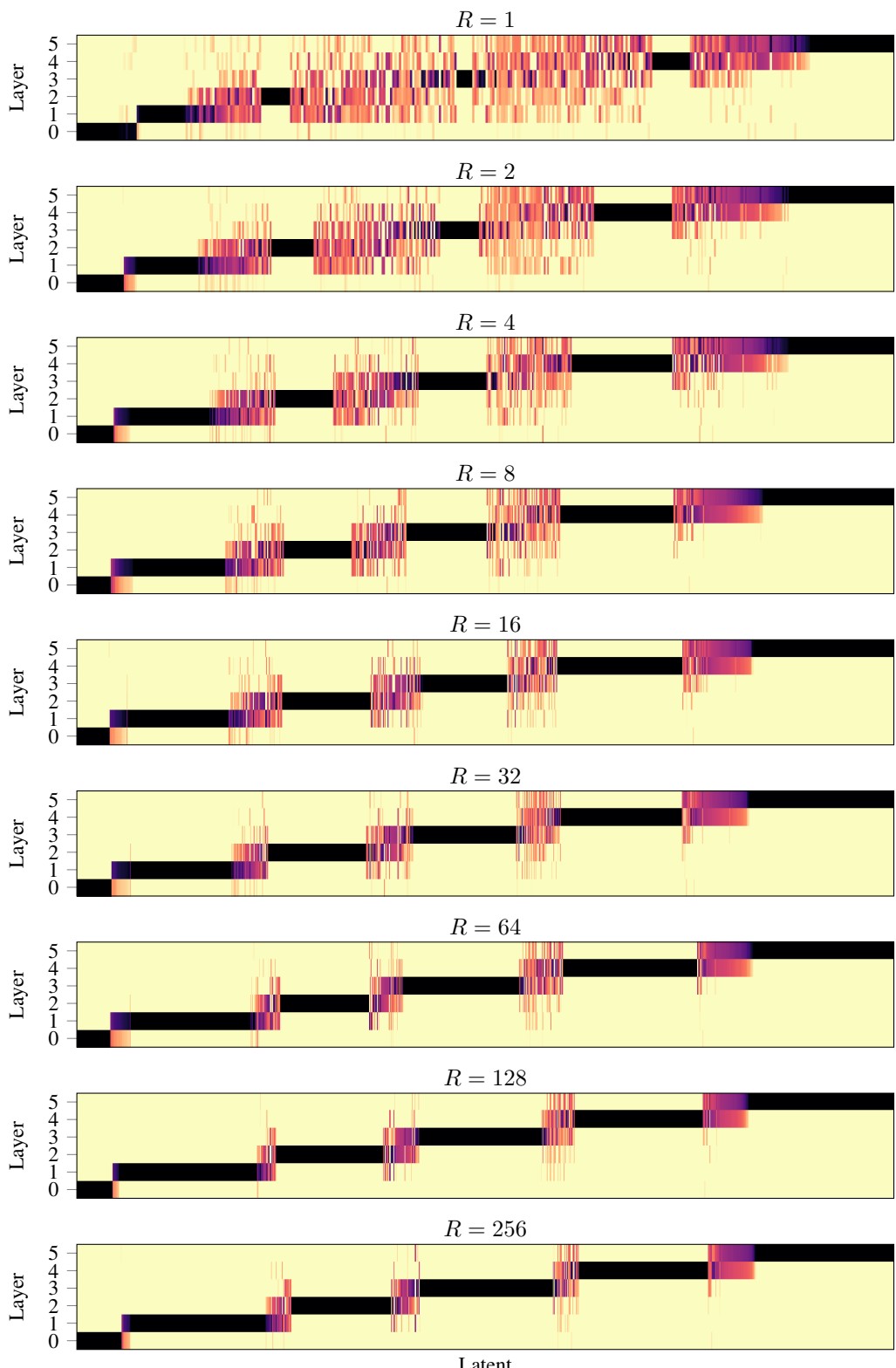

Figure 31: Heatmaps of the distributions of latent activations over layers for a single example prompt. Here, we plot the distributions for MLSAEs trained on Pythia-70m with sparsity $k = 32$. The example prompt is "When John and Mary went to the store, John gave" (Wang et al., 2022). We provide further details in Figure 3.

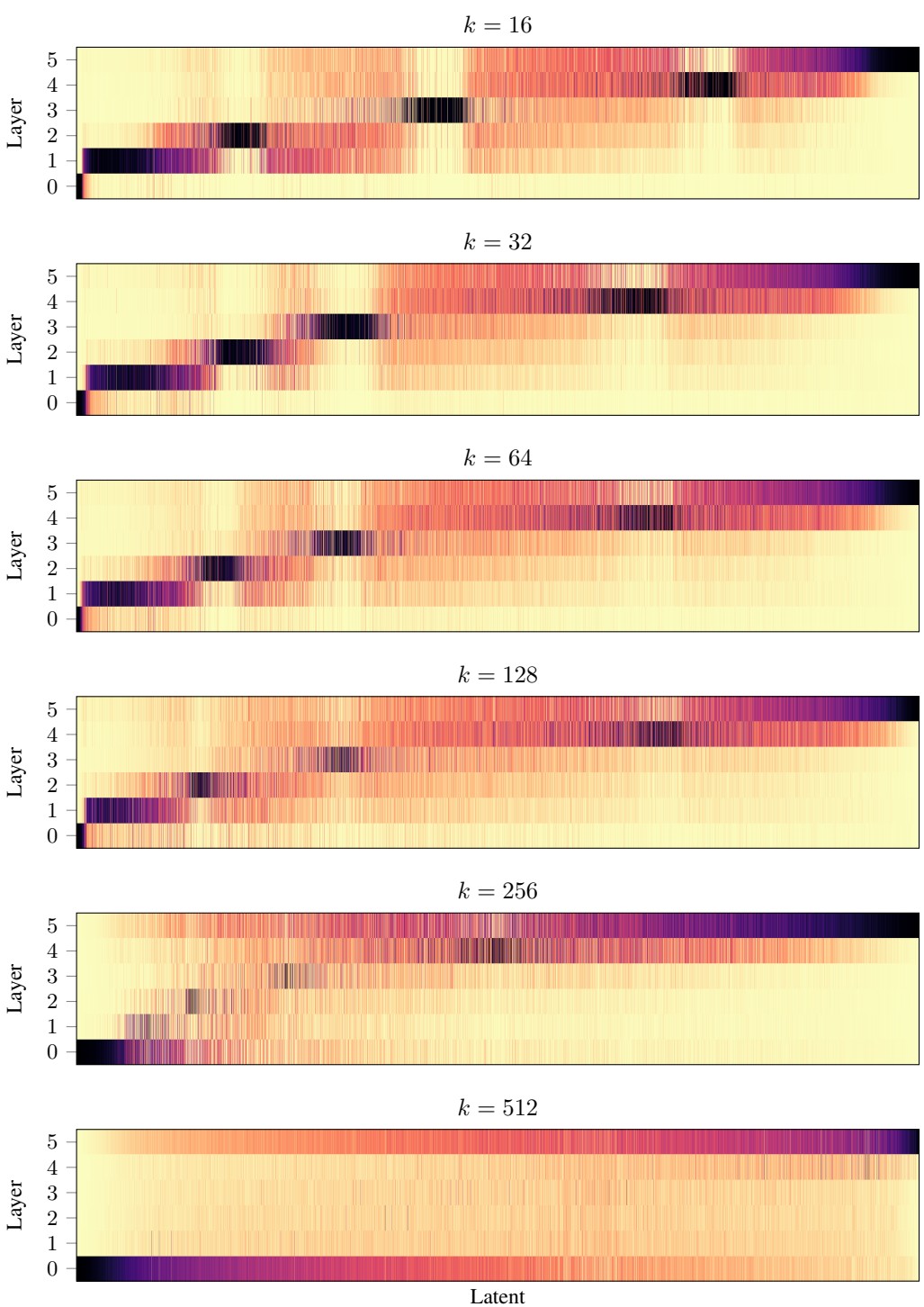

Figure 32: Heatmaps of the distributions of latent activations over layers when aggregating over 10 million tokens from the test set. Here, we plot the distributions for MLSAEs trained on Pythia-70m with an expansion factor of $R = 64$. We provide further details in Figure 2.

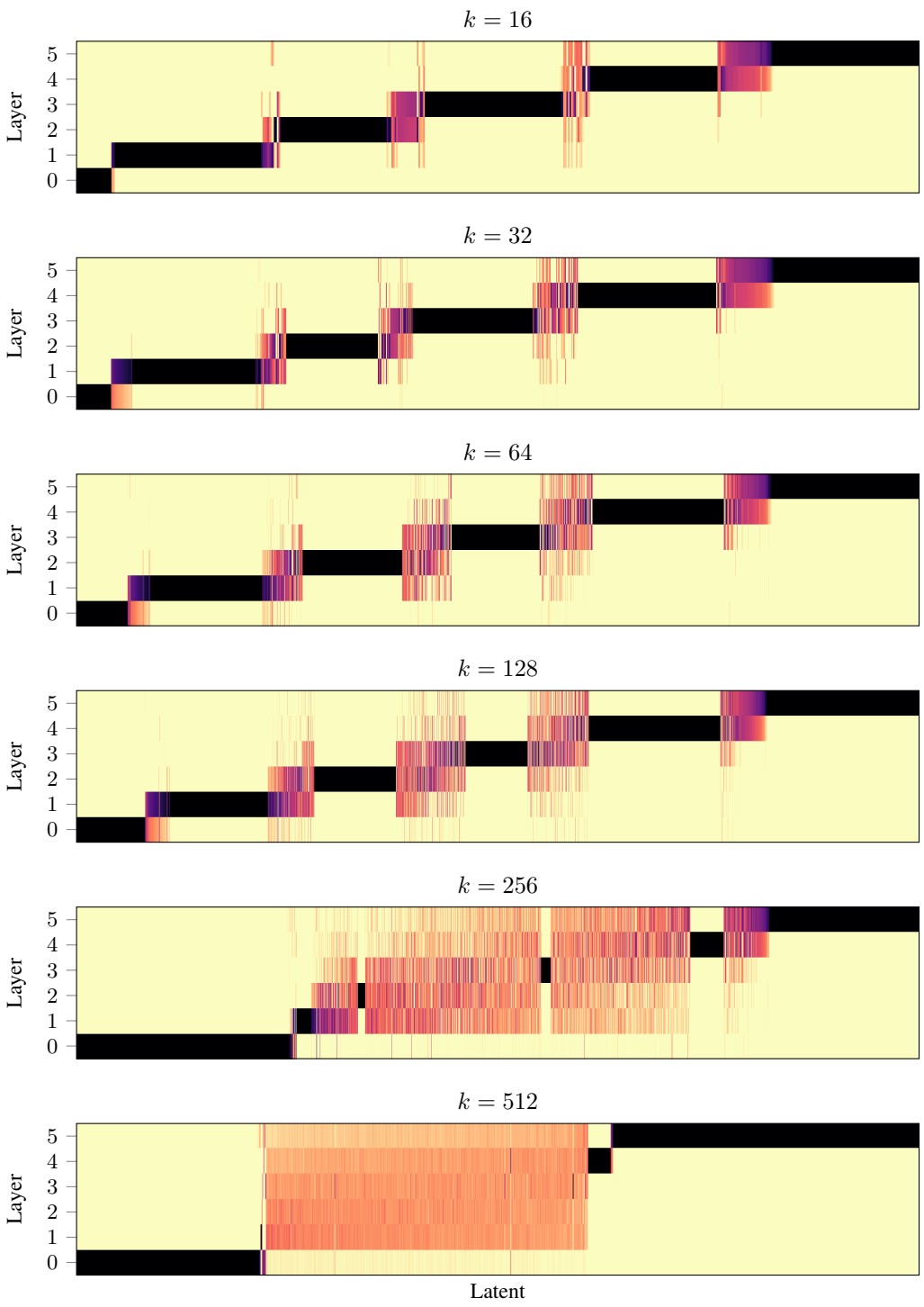

Figure 33: Heatmaps of the distributions of latent activations over layers for a single example prompt. Here, we plot the distributions for MLSAEs trained on Pythia-70m with an expansion factor of $R = 64$. The example prompt is "When John and Mary went to the store, John gave" (Wang et al., 2022). We provide further details in Figure 3.

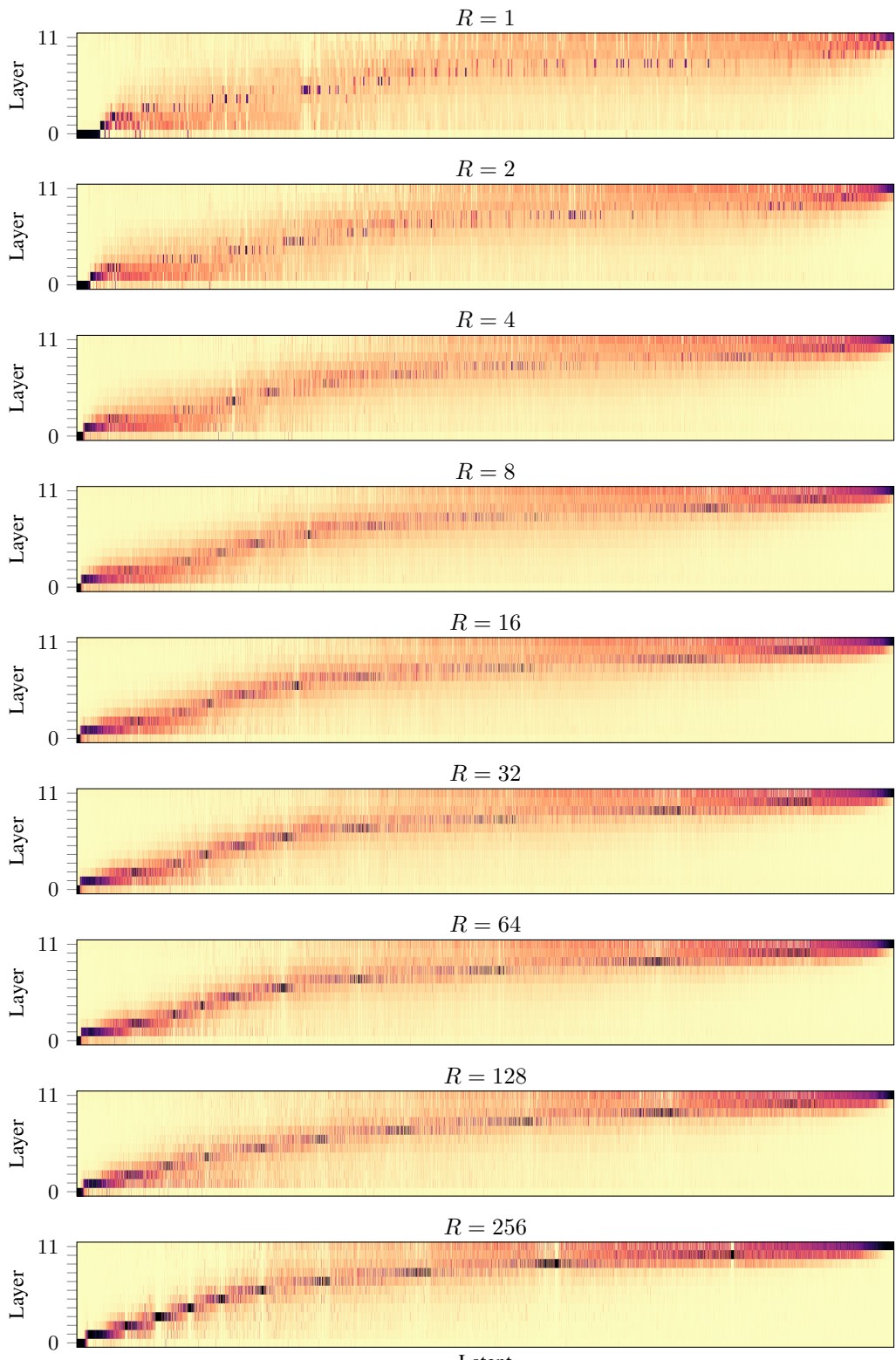

Figure 34: Heatmaps of the distributions of latent activations over layers when aggregating over 10 million tokens from the test set. Here, we plot the distributions for MLSAEs trained on Pythia-160m with sparsity $k = 32$. We provide further details in Figure 2.

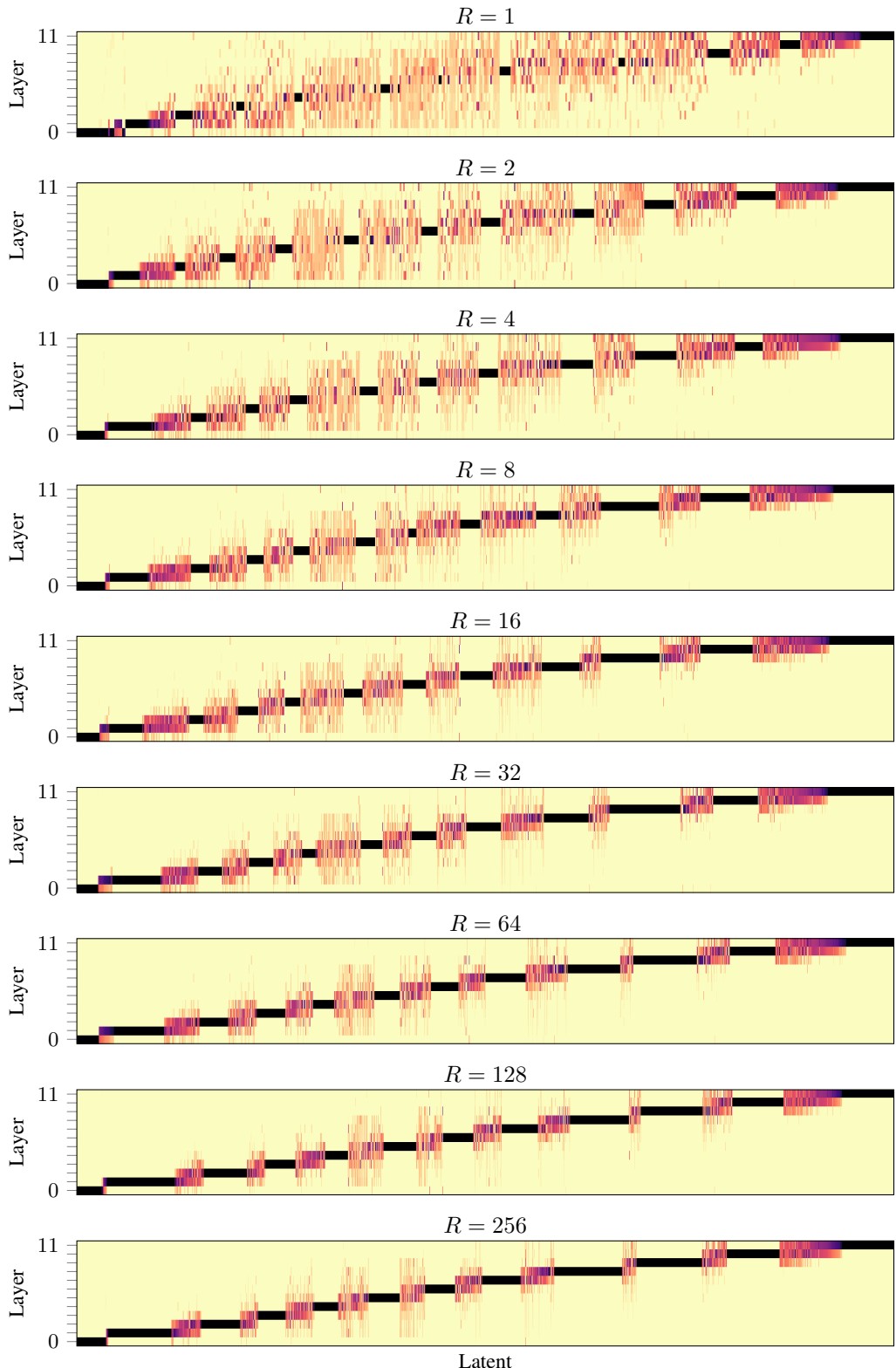

Figure 35: Heatmaps of the distributions of latent activations over layers for a single example prompt. Here, we plot the distributions for MLSAEs trained on Pythia-160m with sparsity $k = 32$. The example prompt is "When John and Mary went to the store, John gave" (Wang et al., 2022). We provide further details in Figure 3.

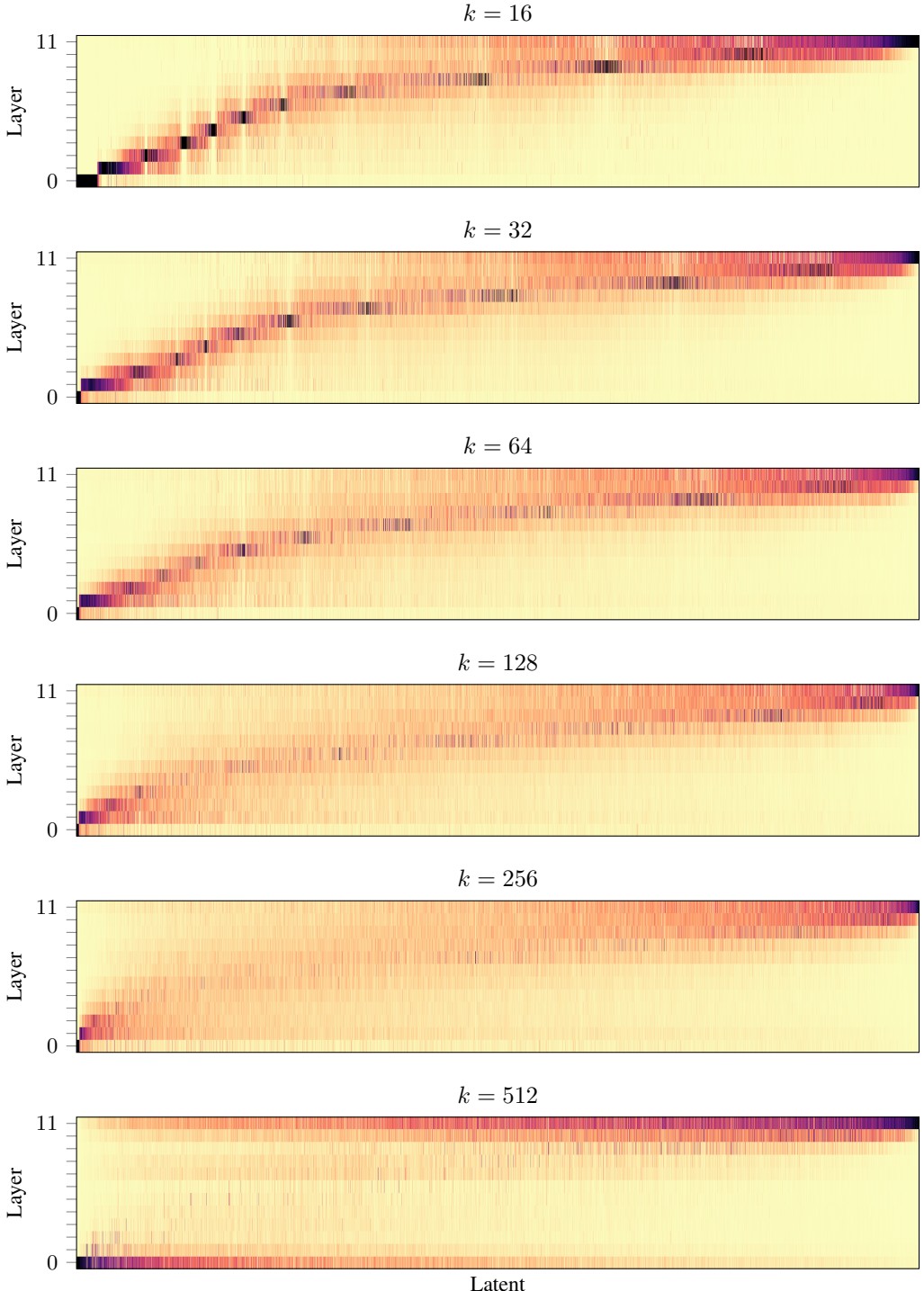

Figure 36: Heatmaps of the distributions of latent activations over layers when aggregating over 10 million tokens from the test set. Here, we plot the distributions for MLSAEs trained on Pythia-160m with an expansion factor of $R = 64$. We provide further details in Figure 2.

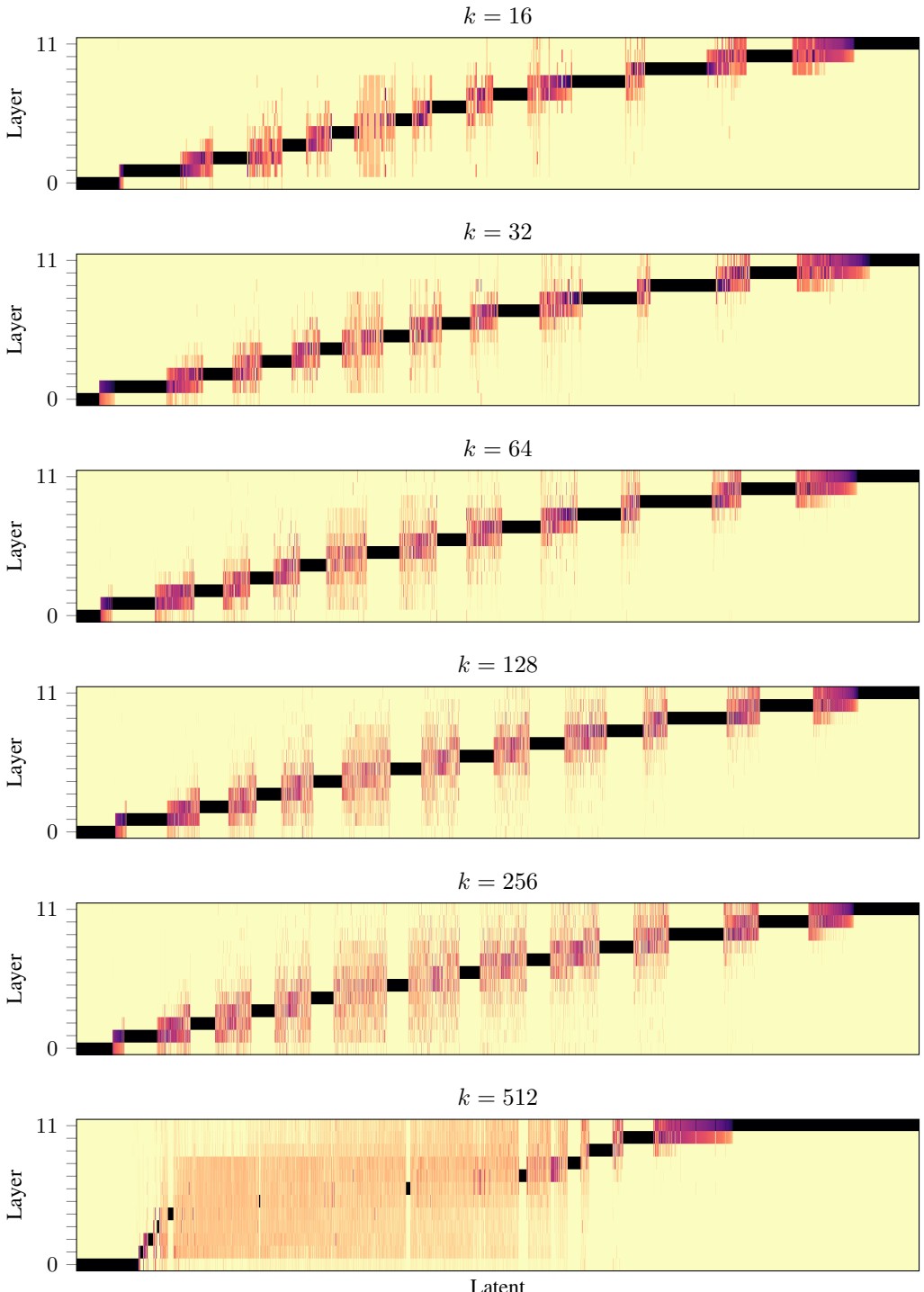

Figure 37: Heatmaps of the distributions of latent activations over layers for a single example prompt. Here, we plot the distributions for MLSAEs trained on Pythia-160m with an expansion factor of $R = 64$. The example prompt is "When John and Mary went to the store, John gave" (Wang et al., 2022). We provide further details in Figure 3.

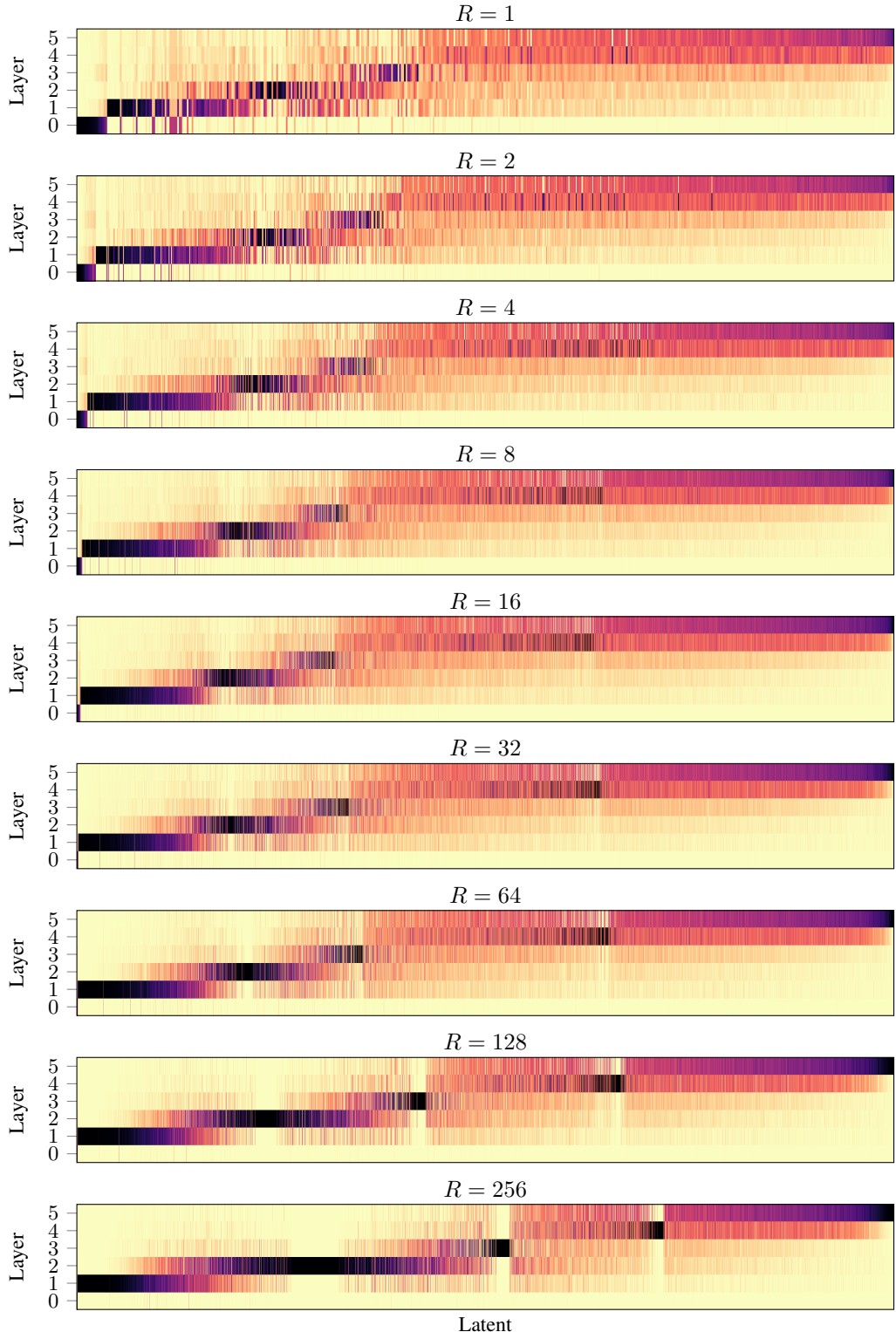

Figure 38: Heatmaps of the distributions of latent activations over layers when aggregating over 10 million tokens from the test set. Here, we plot the distributions for tuned-lens MLSAEs trained on Pythia-70m with sparsity $k = 32$. We provide further details in Figure 2.

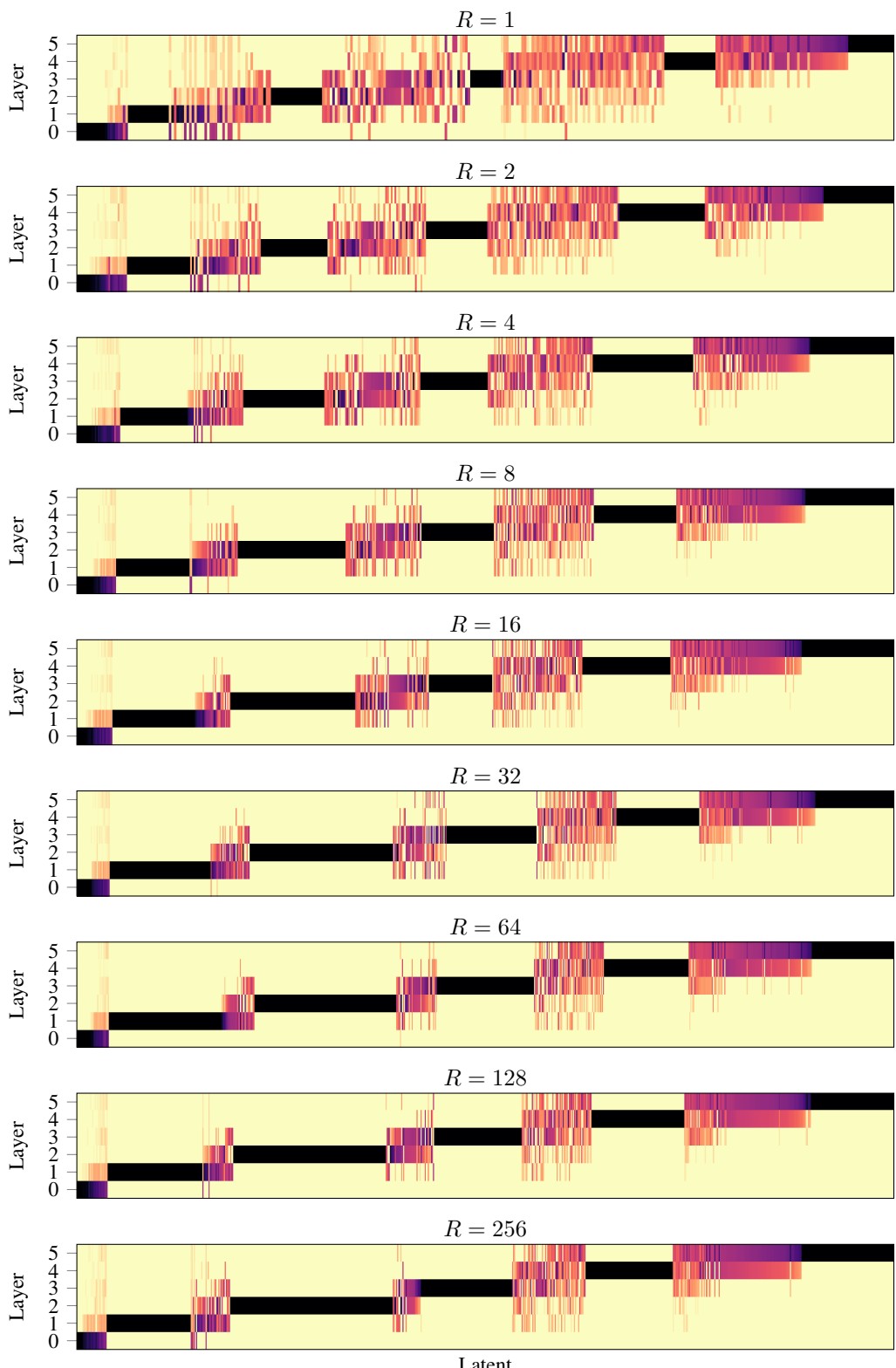

Figure 39: Heatmaps of the distributions of latent activations over layers for a single example prompt. Here, we plot the distributions for tuned-lens MLSAEs trained on Pythia-70m with sparsity $k = 32$. The example prompt is "When John and Mary went to the store, John gave" (Wang et al., 2022). We provide further details in Figure 3.

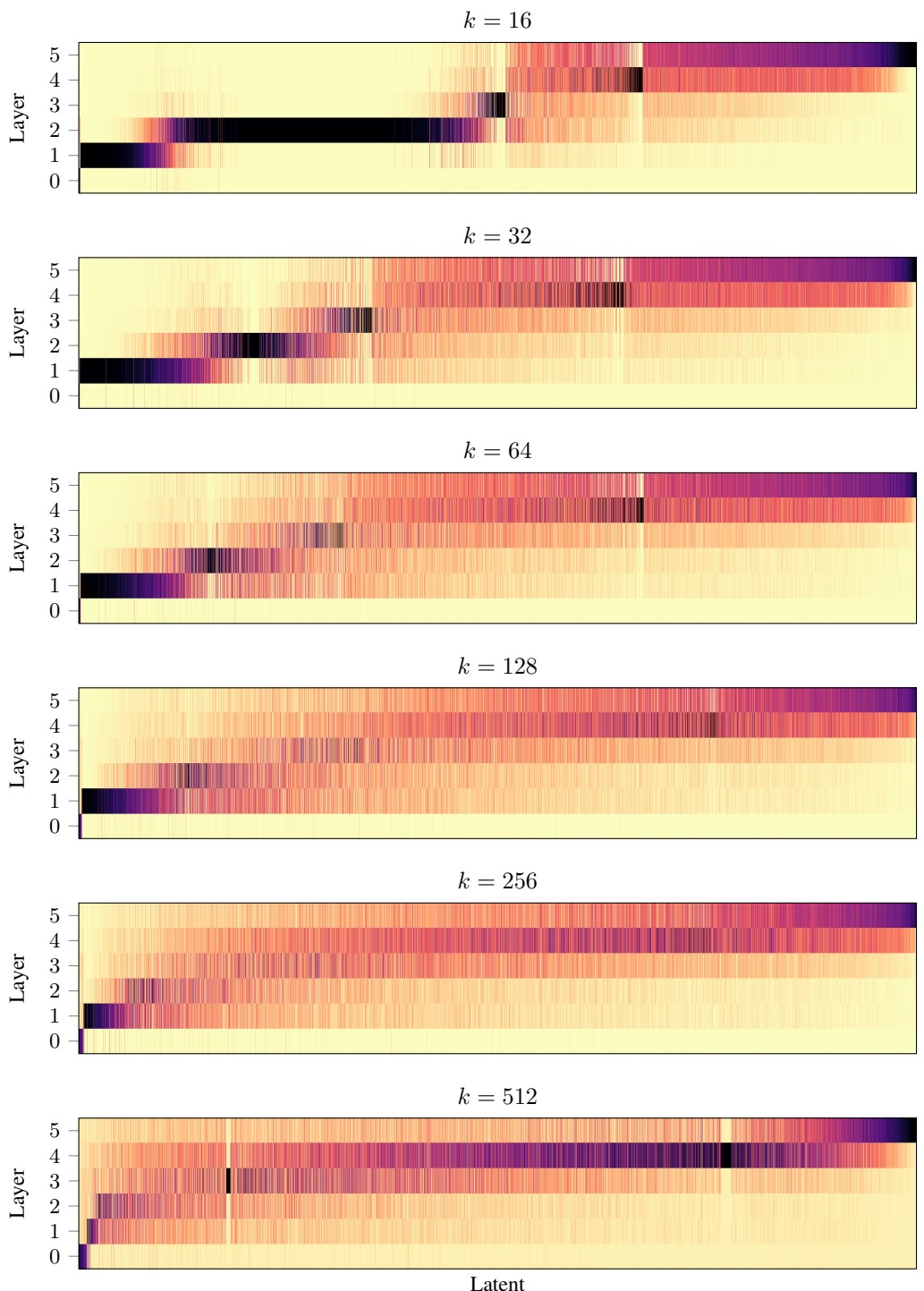

Figure 40: Heatmaps of the distributions of latent activations over layers when aggregating over 10 million tokens from the test set. Here, we plot the distributions for tuned-lens MLSAEs trained on Pythia-70m with an expansion factor of $R = 64$. We provide further details in Figure 2.

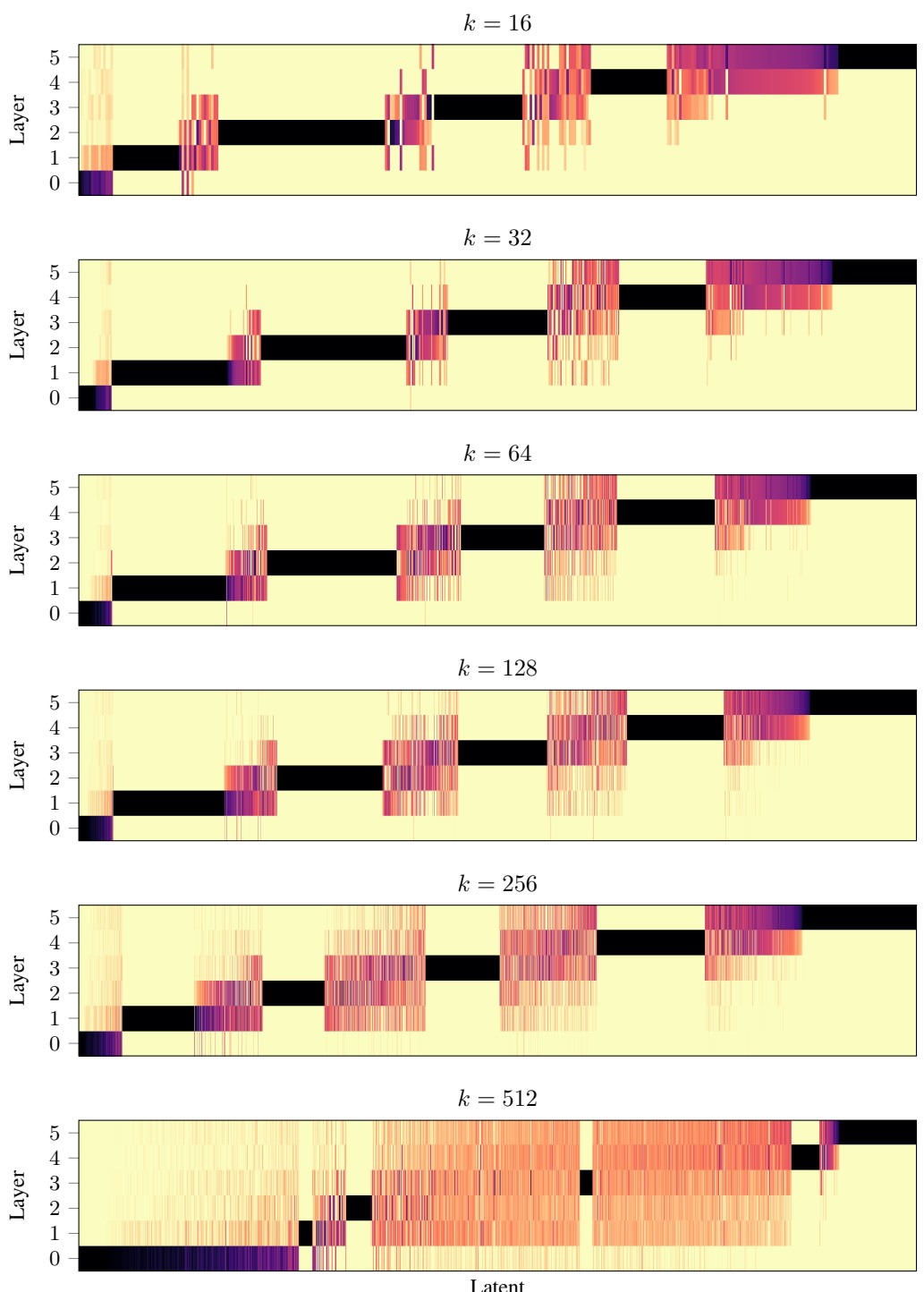

Figure 41: Heatmaps of the distributions of latent activations over layers for a single example prompt. Here, we plot the distributions for tuned-lens MLSAEs trained on Pythia-70m with an expansion factor of $R = 64$. The example prompt is "When John and Mary went to the store, John gave" (Wang et al., 2022). We provide further details in Figure 3.

