# OpenReview forum: "Residual Stream Analysis with Multi-Layer SAEs"
_ICLR.cc/2025/Conference — ICLR 2025 Poster_

### Official Review · Reviewer_d1f6 · 2024-10-17

**Soundness:** 3
**Presentation:** 3
**Contribution:** 4
**Rating:** 6
**Confidence:** 4

**Summary:**

This paper introduces a novel technique for training a SAE which can be applied to any layer of a model, called a MLSAE, by using training samples drawn from all layers of the model. This is beneficial because it is desirable to be able to see how the active latents change from layer to layer. To address differences in norm between layers, the paper normalizes activations and optionally applies tuned lens to help ensure that activations share the same basis. The paper studies to what extent latents are active across multiple layers as a function of model size, SAE sparsity, and SAE width using Pythia models of various sizes.

**Strengths:**

The problem of studying SAE latents across multiple layers is an important and understudied problem, and the contribution of training a single SAE on activations from multiple layers is novel. Using TunedLens as a pre-processing step is also novel. The paper is clear and understandable.

**Weaknesses:**

The paper does not evaluate the quality of the resulting MLSAEs using standard metrics, such as variance explained and cross-entropy loss when the SAE is inserted into the model. It is very easy to train a poor quality SAE, and it is not clear if the MLSAEs in the paper are decent SAEs by these metrics. There is also no performance comparison to a standard single-layer SAEs to see if MLSAEs result in a performance decrease when evaluated on a single layer. The paper only evaluates on Pythia models and topk SAEs - it would be beneficial to study other models and SAE architectures if possible.

**Questions:**

- The paper mentions not having a pre-trained TunedLens in some cases - where are these pre-trained TunedLenses from?
- Why does Figure 6 have only Pythia 70m and 160m, and only TunedLens for 70m? The rest of the paper has many more sizes evaluated with and without TunedLens
- I was surprised that in Figure 3 so many latents are active at only a single layer. Is it possible the MLSAE has effectively learned the same latent multiple times for different layers?

---

> ### Author Response · Authors · 2024-11-25
> **Response to Reviewer d1f6**
>
> Thanks for your positive comments!
>
> > The paper does not evaluate the quality of the resulting MLSAEs using standard metrics, such as variance explained and cross-entropy loss when the SAE is inserted into the model. It is very easy to train a poor-quality SAE, and it is not clear if the MLSAEs in the paper are decent SAEs by these metrics.
>
> Thank you for pointing this out. While we included such an evaluation in Appendix B of the original manuscript, we agree that we did not signpost this clearly in the main text (previously L079–080). As such, in the revised paper, we added a top-level summary of our evaluation results to Section 4.1 of the main text as Table 1 (p. 6). Specifically, we report the FVU and MSE reconstruction errors, the $L^1$ norm, and the increase in the cross-entropy loss and KL divergence when we replace the residual stream activations at a given layer by their reconstruction (‘downstream loss’ metrics, following Gao et al., 2024).
>
> > There is also no performance comparison to standard single-layer SAEs to see if MLSAEs result in a performance decrease when evaluated on a single layer.
>
> Thank you for pointing this out. In Appendix B of the original manuscript, we compared the results of our evaluation to single-layer SAEs trained by Marks et al. (2024), Gao et al. (2024), Braun et al. (2024), and Lieberum et al. (2024). Specifically, we found that the FVU (normalized) reconstruction error and increase in cross-entropy loss were comparable. However, we agree with Reviewers g44V and d1f6 that further comparison to single-layer SAEs was needed, and we have now trained single-layer SAEs on each layer of Pythia-70m and 160m and evaluated them on data from every layer. This analysis appears in Appendix B.4 and Figures 15–17 in the revised paper. It confirms that, for a given layer of test data, a multi-layer SAE achieves very similar performance to a single-layer SAE trained at that layer and much better performance than a single-layer SAE trained at a different layer. Interestingly, we also find that applying tuned-lens transformations to the input vectors from each layer during training and evaluation does not improve the ability of single-layer SAEs to generalize to test data from other layers.
>
> >The paper only evaluates on Pythia models and topk SAEs - it would be beneficial to study other models and SAE architectures if possible.
>
> We chose the Pythia suite of transformers because it is highly reproducible, as noted by Reviewer g44V, and offers a controlled setting to study how our results scale across model sizes. We chose TopK SAEs due to the simplicity of directly controlling the sparsity ($L^0$ norm) of the latent space (rather than indirectly, by tuning the $L^1$ penalty). However, we agree that confirming our findings for other transformers and SAE architectures would strengthen the paper, so we have now trained a multi-layer TopK SAE with our default hyperparameters on GPT-2 small (similar in size to Pythia-160m). In the revised paper, the results for GPT-2 small are described in Appendix B.3, along with aggregate and single-prompt heatmaps of the distributions of latent activations over layers (Figures 13 and 14). The qualitative results are similar to the Pythia models (e.g., Figures 2 and 3).
>
> > The paper mentions not having pre-trained TunedLens in some cases - where are these pre-trained TunedLenses from?
>
> The reviewer rightly points out that we should have included the source of the pre-trained lenses we use in the paper: the HuggingFace repository https://huggingface.co/spaces/AlignmentResearch/tuned-lens. The authors of Belrose et al. (2023) and the implementation we describe in Section 3.2 made these models publicly available. We have added a note to clarify this point in Section 3.3 (L297–298) of the revised paper.
>
> > Why does Figure 6 have only Pythia 70m and 160m, and only TunedLens for 70m? The rest of the paper has many more sizes evaluated with and without TunedLens
>
> Thank you for pointing this out. Most figures in the main text describe MLSAEs trained on different models with fixed hyperparameters (an expansion factor of $R=64$ and sparsity $k=32$). In contrast, many figures in the appendix vary the expansion factor $R$ between 1 and 256 and the sparsity $k$ between 16 and 512 for the smallest models (Pythia-70m and 160m). In the original manuscript, we intended Figure 6 to demonstrate the trends in the variance ratios when varying these hyperparameters. Unfortunately, it was prohibitively computationally expensive to perform the same hyperparameter sweeps for all models. In the revised paper, we have moved Figure 6 to the appendix; it is now Figure 22.

---

> ### Author Response · Authors · 2024-11-25
> **Response 2 to Reviewer d1f6**
>
> > I was surprised that in Figure 3 so many latents are active at only a single layer. Is it possible the MLSAE has effectively learned the same latent multiple times for different layers?
>
> Thank you for calling attention to this. In the original manuscript, we included an analysis of the ‘mean max cosine similarity’ between decoder weight vectors (Sharkey et al., 2022) in Appendix B.3 and Figure 13 as a preliminary measure of the pairwise similarities between latents, and the results did not suggest a large number of ‘duplicate’ latents. However, we agree that this point merits further investigation.
>
> In the revised paper, we have added comparisons between the distribution of pairwise cosine similarities between decoder weight vectors and (a) an equal number of independent and identically normally distributed vectors of equal length as a negative control and (b) a smaller number of i.i.d. normal vectors copied for each model layer plus a small amount of noise. Figure 19 shows that the distributions for decoder weight vectors are slightly heavier-tailed and right-shifted, i.e., a pair of MLSAE latents are slightly more likely to have high cosine similarity than a pair of i.i.d. normal vectors. However, we do not find evidence for many pairs of latents that are highly similar but active at different layers.

---

> ### Author Response · Authors · 2024-11-25
> **References mentioned in response to Reviewer d1f6**
>
> Belrose et al., 2023: https://arxiv.org/abs/2303.08112
>
> Braun et al., 2024: https://arxiv.org/abs/2405.12241
>
> Gao et al., 2024: https://arxiv.org/abs/2406.04093
>
> Lieberum et al., 2024: https://arxiv.org/abs/2408.05147
>
> Marks et al., 2024: https://arxiv.org/abs/2403.19647
>
> Sharkey et al., 2022: https://www.alignmentforum.org/posts/z6QQJbtpkEAX3Aojj/interim-research-report-taking-features-out-of-superposition

---

> ### Author Response · Authors · 2024-12-02
> **Final message**
>
> We were wondering whether you had time to consider our response?
>
> In short, we agree that missing quality metrics such as variance explained and cross-entropy loss would be a worrying oversight.  However, these quality metrics were available in Appendix B in the original paper, but we accept that that wasn't flagged clearly enough.  We have therefore expanded Appendix B, and included a summary of these results in the main text of the revised manuscript.
>
> Given the clarifications in our response, along with your otherwise excellent comments ("an important and understudied problem, and the contribution of training a single SAE on activations from multiple layers is novel") and ratings:
> > Soundness: 3: good
>
> > Presentation: 3: good
>
> > Contribution: 4: excellent
>
> Would consider raising your score to an accept level?

---

> > ### Comment · Reviewer_d1f6 · 2024-12-02
> >
> > Thank you for the clarifications and for pointing out that the MLSAE does not result in dramatically worse performance compared to single layer SAEs. I've updated my score.

---

### Official Review · Reviewer_F8UN · 2024-10-22

**Soundness:** 3
**Presentation:** 3
**Contribution:** 2
**Rating:** 6
**Confidence:** 2

**Summary:**

This study introduces a novel approach called MLSAE for analyzing the evolution of representations across various layers in Transformers. The author presents comprehensive experimental evidence to elucidate the statistical characteristics of representations at different layers. Nonetheless, how these statistical analyses can help the understanding of the inherent mechanisms within Transformers not that clear.

**Strengths:**

1. This work proposes the MLSAE method to analyze the interpretability of information transmitted through residual flow in Transformers.
2. The author only uses one SAE in multi-layer Transformers.
3. The author analyzes the fraction and variance of latents of models of different sizes through detailed experiments.

**Weaknesses:**

1. As indicated in both the Abstract and Conclusion sections, the author investigates the transformation of representations across various layers within Transformers. However, the analysis is limited to a statistical perspective, focusing solely on the fractions and variances of representations at different layers. The linkage between this statistical overview and a deeper understanding of the underlying mechanisms of Transformers is not explicitly articulated. It remains unclear to me how these statistical descriptions facilitate insight into the working mechanism of Transformers.
2. The motivation for the author's selection of Stacked Autoencoders (SAE) as the dimensionality reduction technique is unclear for me. The advantages of employing an SAE, which necessitates training, over more traditional statistical methods such as Principal Component Analysis (PCA) or t-distributed Stochastic Neighbor Embedding (t-SNE) for analyzing the latent space, are not clearly delineated. This choice warrants further clarification.

**Questions:**

Please refer to the Weakness Section.

---

> ### Author Response · Authors · 2024-11-25
> **Response to Reviewer F8UN**
>
> Thanks for your comments!
>
> > It remains unclear to me how these statistical descriptions facilitate insight into the working mechanism of Transformers.
>
> Please see our global response above for our understanding of our mechanistic contributions.
>
> > The motivation for the author's selection of Stacked Autoencoders (SAE) as the dimensionality reduction technique is unclear for me.
>
> Importantly, SAE stands for "Sparse Autoencoders," not "Stacked Autoencoders," and SAEs are not a dimensionality reduction technique. Instead, SAEs learn a higher-dimensional or "overcomplete" basis. In the context of language-model interpretability, the use of SAEs starts from the observation that the dimension of the latent vectors in LLMs is relatively small (e.g., 1024), given that LLMs can represent, in some sense, all possible token "meanings." This observation motivated the hypothesis that we can understand LLM activations as lower-dimensional projections of a sparsely activated, higher-dimensional representation space whose basis vectors are easier for humans to interpret. This hypothesis has been explored in depth by, e.g., Elhage et al., 2022; Park et al., 2023. Critically, the resulting interpretable, sparse basis is overcomplete: it has more dimensions than the original LLM activation vectors.
>
> > The advantages of employing an SAE, which necessitates training, over more traditional statistical methods such as Principal Component Analysis (PCA) or t-distributed Stochastic Neighbor Embedding (t-SNE) for analyzing the latent space, are not clearly delineated. This choice warrants further clarification.
>
> As described above, the goal is to find an overcomplete basis in which the original LLM activation vectors become sparser and more interpretable. In contrast, PCA and t-SNE give a basis smaller than (or the same size as) the input vectors and do not promote sparsity. Indeed, t-SNE is typically used to visualize high-dimensional data by embedding it into two or three dimensions and prioritizes visual separation over the interpretability of the embedded dimensions. While researchers have used techniques like t-SNE and UMAP to analyze the representations learned by SAEs (e.g., Gao et al., 2024; Appendix E.7), we cannot use them to find higher-dimensional, sparse, interpretable representations.
>
> Importantly, there is a classical algorithm with strong practical and theoretical connections to SAEs: independent component analysis (ICA). We can understand SAEs as ICA with a noise model optimized by gradient descent. We note that Cunningham et al. (2023) compared PCA, ICA, and SAEs in the context of language-model interpretability, finding that SAEs produced more interpretable directions by their quantitative measures. Finally, it is important to note that ICA also requires iterative optimization or training (e.g., Bell and Sejnowski 1995, Sec. 2.4). In the revised paper, we have expanded on the connection between ICA and SAEs on L092–096.
>
> In summary, the choice of SAEs was motivated by their popularity and apparent effectiveness for learning interpretable, higher-dimensional representations for LLM activation vectors, but one could adapt the key idea with our method to other algorithms.

---

> ### Author Response · Authors · 2024-11-25
> **References mentioned in response to Reviewer F8UN**
>
> Bell and Sejnowski, 1995: https://pubmed.ncbi.nlm.nih.gov/7584893/
>
> Cunningham et al., 2023: https://arxiv.org/abs/2309.08600
>
> Elhage et al., 2022: https://arxiv.org/abs/2209.10652
>
> Gao et al., 2024: https://arxiv.org/abs/2406.04093
>
> Park et al., 2023: https://arxiv.org/abs/2311.03658

---

> > ### Comment · Reviewer_F8UN · 2024-11-27
> >
> > Thank you for your clarification.
> >
> > According to the global response, I believe your work has a great contribution to the community. I would like to raise my score.
> >
> > However, similar to the Weakness mentioned by Reviewer d1f6,
> >
> > > it would be beneficial to study other models and SAE architectures if possible.
> >
> > The experiments shown in this work only conduct on really small model (the largest one is 1B). However, such small models are rarely used in practical scenarios.
> >
> > I am not sure if your findings still holds for the models really used in practical scenarios? To the best of my knowledge, the smallest model used in practical scenarios contains approximate 3B parameter (https://huggingface.co/openbmb/MiniCPM-V). I believe experiments on practical models can enhance this paper a lot. And I am really looking forward to hearing your opinion on this question.

---

> ### Author Response · Authors · 2024-12-02
> **Response to Reviewer F8UN**
>
> Thank you for carefully considering our response!
>
> In response to your most recent comment, we trained a multi-layer SAE on Llama-3.2-3B, which gave qualitatively very similar results to those shown in the paper.
>
> This is expected: qualitative findings in mechanistic interpretability (particularly those derived using SAEs) have historically generalized remarkably well across model sizes. For example, the initial SAE papers (Cunningham et al., 2023; Bricken et al., 2023) used much smaller models than we do, and yet their qualitative findings were subsequently replicated in the largest commercial models (Gao et al., 2024; Templeton et al., 2024).
>
> We plan to additionally run a few 7b or 8b models for the camera ready.
>
> Unfortunately, OpenReview does not currently allow revisions. As such, we will, of course, put these results in the camera-ready. We hope you are excited to see them! Given that, and given your kind statement that "I believe your work has a great contribution to the community", we were wondering whether you would consider further raising your score to accept?

---

### Official Review · Reviewer_WQNj · 2024-11-03

**Soundness:** 3
**Presentation:** 3
**Contribution:** 3
**Rating:** 6
**Confidence:** 4

**Summary:**

The paper proposes a method based on sparse autoencoders (SAEs) for recovering features and analyzing the distributions of activations of the same features (SAE latents) across layers in Pythia language transformers of different sizes. The SAE is trained in a manner that treats per-token feature vectors (potentially transformed with tuned lens) from all layers as independent training examples. Hence, the method is named Multi-layer SAE.

Contrary to the authors' hypothesis that features often remain active at later layers once they are activated, the experiments show activations of the latents are often more localized and active at a single layer.

**Strengths:**

- The paper is well written.
- The experimental evaluation is well done.
- The research area of the paper is mechanistic interpretability, which is a potentially very useful for understanding complex models and important for AI safety.

**Weaknesses:**

1. Some things might be unclear (on the first reading):
	- this layer may differ for different tokens or prompts -> which layer it is might differ ... ?
	- Under multiple figures: "expected value of the layer" -> "average layer index" or "expected value of the layer index"
	- Eq. (3): The meaning of $\operatorname{Var}$ could be clarified.
	- In some places, the notation is a bit inconsistent or non-standard.
		- In section 3.1, $\bf x\in\mathbb R^d$, but in Eq. (3), $\bf x$ represents a sequence of vectors $\bf x_\ell$.
		- Eq. (9): $\hat{\bf x}_\ell$ is the same as $\bf x_\ell$. It is a bit confusing that two symbols are used for the same thing.
2. Some training details seem missing: How are layers for mini-batches sampled?
3. It is not very clear to me how the paper is advancing the research in mechanistic interpretability. Perhaps commenting more on what motivated the investigations and what the results suggest for future work directions would be valuable.
4. The paper does not state whether the source code for reproducing the experiments will be published.

There are a few errors:
- L245: "ratio between the model dimension and the number of latents" should be the other way around.
- L353: Should $\mathbb E_T[L\mid J=j]$ be $\mathbb E[L\mid J=j]$ (or, equvalently, $\mathbb E_L[L\mid J=j]$)?

I apologize for any errors on my part.

**Questions:**

Questions:
1. How much of computational resources did the experiments require?
2. L301: "we found the mean cosine similarities increased accordingly". This is not true for smaller models. Do you have any intuitions about that?
3. The analysis is normalizing histograms of activations over layers to get distributions. This loses information about relative frequencies of activations of different latents or about the number of layers in which each latent is active. Do you have any comments on this?
4. The paper uses metrics based on variance of the layer index. When looking at a single token, they depend on the number of activated layers as well as their distance. Could measuring the number of activated layers be more useful or easy to interpret for some purposes?
5. What is the intuition behind the hypothesis that features often remain active at later layers once they are activated? Is the hypothesis based on analogous observations when the SAE is trained only on the last layer?

If the questions make sense, I would like the final paper to address them.

Suggestions:
1. Eq (8): Note that the residual connection matters only because of L2 regularization (and because if it effectively changes the initialization).

---

> ### Author Response · Authors · 2024-11-25
> **Response to Reviewer WQNj**
>
> Thanks for your positive comments and detailed suggestions!
>
> > this layer may differ for different tokens or prompts -> which layer it is might differ ...?
>
> Thanks for pointing this out. We have changed this sentence on L019–020 in the revised paper to read: "Interestingly, we find that individual latents are often active at a single layer for a given token or prompt, but the layer at which the latent is active may differ for different tokens or prompts."
>
> > Under multiple figures: "expected value of the layer" -> "average layer index" or "expected value of the layer index"
>
> Thanks! In the revised paper, we have replaced the phrase “expected value of the layer”  with “expected value of the layer index” on L133, L161, and L370.
>
> > Eq. (3): The meaning of $\operatorname{Var}$ could be clarified.
>
> Thanks! We have clarified this below Eq. 3 with the sentence, “Here, $\operatorname{Var}$ is the variance.”
>
> > In section 3.1, $\bf x \in \mathbb{R}^d$, but in Eq. (3), $\bf x$ represents a sequence of vectors $\bf x_\ell$.
>
> Thanks! Our use of subscripts to indicate the layer index of activation vectors needed to be clarified. We have removed the layer indices from the subscripts of vectors (e.g., $\bf x_\ell$) in Eqs. 3, 8, and 9 because they are unnecessary to convey the meaning.
>
> > Eq. (9): $\hat{\bf x}\ell$ is the same as $\bf x\ell$. It is a bit confusing that two symbols are used for the same thing.
>
> Thanks! We have added a sentence to explain our notation after Eq. 9: $\hat{\bf x}$ is the vector output by the autoencoder, i.e., the reconstruction, whereas $\bf x$ is the vector input to the autoencoder. Both the output and input vectors have the same shape but are not equal. As mentioned above, we have removed the layer indices from Eqs. 3, 8, and 9.
>
> > Some training details seem missing: How are layers for mini-batches sampled?
>
> Thanks for pointing this out. We have added an explanation to Appendix A in the revised paper (L764–766): we construct a batch of input activation vectors by performing the forward pass of the transformer over a sequence of tokens, collecting the residual stream activation vectors from every layer, and stacking them together. Hence, each batch of activations has an equal number of vectors from each layer of the transformer (the number of tokens in the sequence).
>
> > It is not very clear to me how the paper is advancing the research in mechanistic interpretability. Perhaps commenting more on what motivated the investigations and what the results suggest for future work directions would be valuable.
>
> Please see our global response above for our understanding of our mechanistic contributions.
>
> > The paper does not state whether the source code for reproducing the experiments will be published.
>
> We have included anonymized source code as supplementary material. We will provide a more convenient link in the camera-ready.
>
> > L245: “ratio between the model dimension and the number of latents” should be the other way around.
>
> Fixed!
>
> > L353: Should $E_T[L|J=j]$ be $E[L|J=j]$ (or, equivalently, $E_L[L|J=j]$?
>
> Fixed!
>
> > How much of computational resources did the experiments require?
>
> Thanks for pointing this out. The revised paper describes the computational cost of training an MLSAE at the end of Section 3.2 (L241–245). In brief, we ran most experiments on a single NVIDIA GeForce RTX 3090 GPU for between 12 and 24 hours; we ran the largest experiments (e.g., with Pythia-1b or an expansion factor of $R=256$) on a single NVIDIA A100 80GB GPU for up to three days.
>
> > L301: “we found the mean cosine similarities increased accordingly.” This is not true for smaller models. Do you have any intuitions about that?
>
> Thanks for pointing out our confusing phrasing. We intended by this statement to claim that the mean cosine similarity between residual stream activation vectors increases as the size of the underlying transformer increases, which is true in Figure 1. However, we agree that the phrase "the mean cosine similarities increased accordingly" is unclear: one could take it to mean that the mean cosine similarity increased across layers. In Figure 1, this is indeed false for smaller models. We have clarified this statement, replacing it with "we found the mean cosine similarities increased as the model size increased" (L346).

---

> > ### Comment · Reviewer_WQNj · 2024-11-27
> >
> > Thank you for the clarifications! I apologize for my error regarding Eq. (9) and other confusions.
> >
> > > > Eq. (3): The meaning of Var could be clarified.
> > > Thanks! We have clarified this below Eq. 3 with the sentence, “Here, Var is the variance.”
> >
> > I apologize for not having been more clear. Perhaps this is a bit nitpicky, but I was confused. The variance is usually considered as a function $\operatorname{Var}\colon (\Omega\to \mathbb R) \to \mathbb R$ whose input is a random variable (we usually implicitly asume a probability space $(\Omega,\Sigma,P)$). However, in Eq. (3), the input $\mathbf x \in \mathbb R^d$ suggests that $\operatorname{Var}$ is a function from $\mathbb R^d \to \mathbb R$ which calculates the variance of a random element of the input $\mathbf x$. If we overload the same symbol for both meanings, $\operatorname{Var}(\mathbf x) \coloneqq \operatorname{Var}(\mathbf x[I])$, where $I\sim \operatorname{Unif}(\{1..d\})$, and the square brackets denote vector indexing.

---

> ### Author Response · Authors · 2024-11-25
> **Response 2 to Reviewer WQNj**
>
> > The analysis is normalizing histograms of activations over layers to get distributions. This loses information about the relative frequencies of activations of different latents or about the number of layers in which each latent is active. Do you have any comments on this?
>
> Thanks for the suggestion! In the revised paper, we added Figures 20 and 21 to Appendix D, which are the equivalent aggregate and single-prompt heatmaps to Figures 2 and 3, except with the un-normalized total latent activations instead of normalized distributions over layers. Surprisingly, the single-prompt heatmap contrasts with the normalized version, suggesting a ‘bimodal’ distribution of activations over layers for some latents (i.e., that some latents are active at non-adjacent layers). Unlike the seemingly unimodal normalized heatmaps, this suggests that latents with relatively large total activations are more likely to be bimodal. We are currently investigating trends in the bimodality of larger latent activations, e.g., how it correlates with the layer index, activation magnitude, etc., and will include these in the camera-ready.
>
> > The paper uses metrics based on the variance of the layer index. When looking at a single token, they depend on the number of activated layers as well as their distance. Could measuring the number of activated layers be more useful or easy to interpret for some purposes?
>
> Thanks for the suggestion! We investigated the number of layers at which each latent is active, which we take to mean that it has a count of non-zero activations above a threshold value (e.g., 10K) over a large sample of input tokens (e.g., 10M). In the revised paper, we have added this analysis to Appendix E.2. This analysis suggests that the mean number of active layers tends to decrease as the number of latents increases relative to the model dimension. Given the preliminary finding of a bimodal structure in the larger latent activations discussed above, it is important to investigate these properties further, and we intend to pursue this for the camera-ready.
>
> > What is the intuition behind the hypothesis that features often remain active at later layers once they are activated? Is the hypothesis based on analogous observations when the SAE is trained only on the last layer?
>
> Our intuition derives from the structure of the network: the residual connection means that the activations from previous layers are maintained, and a layer "writes" new information to the residual stream activations at subsequent layers (Elhage et al., 2021; Ferrando et al., 2024). We also suspected that "deleting" information would be difficult because the network would have to "write" in precisely the opposite direction in the residual stream. Looking at Figure 4, however, we note that a much easier way to effectively "delete" information is to overwhelm it with activation vectors with much greater magnitudes, such that the information written at previous layers becomes negligible.
>
> > Eq (8): Note that the residual connection matters only because of L2 regularization (and because it effectively changes the initialization).
>
> Agreed; we have added a note below Eq. 8 (L265–267) to explain this.

---

> > ### Comment · Reviewer_WQNj · 2024-11-27
> >
> > Thank you for the additional experiments, investigations, and insightful explanations!
> >
> > > Surprisingly, the single-prompt heatmap contrasts with the normalized version, suggesting a ‘bimodal’ distribution of activations over layers for some latents (i.e., that some latents are active at non-adjacent layers). Unlike the seemingly unimodal normalized heatmaps, this suggests that latents with relatively large total activations are more likely to be bimodal.
> >
> > This bimodality looks quite surprising. I couldn't make sense of it for a while. I only expected that some columns will be darker and other lighter. Upon a closer look, it seems to me that individual latents are not as bimodal as it seems at first, but it seems so due to an error in the sorting. Perhaps the sorting value should be divided by the sum over layers?
> >
> > Please also consider adding a colorbar or some other aid so that the values in this figure can be interpreted more accurately.

---

> ### Author Response · Authors · 2024-11-25
> **References mentioned in response to Reviewer WQNj**
>
> Elhage et al., 2021: https://transformer-circuits.pub/2021/framework/index.html
>
> Ferrando et al., 2024: https://arxiv.org/abs/2405.00208

---

> ### Author Response · Authors · 2024-11-27
> **Response**
>
> Thanks for reading the revision so carefully!
>
> We agree about the variance: we've updated the text to read, "Here, Var is the variance, treating x as a random vector, where the randomness is induced by randomizing the token, producing different activation vectors."
>
> We will include all color bars for the camera-ready (there are about 10 figures to update, so I don't believe we're going to be able to finish before the imminent deadline for revisions).
>
> As regards the sorting: good catch!  It does seem there was an error in the sorting.  We have revised Figure 21, and it now looks very similar to the normalised plots (e.g. Figure 3), as we would expect.
>
> We were wondering whether you would consider raising the score to reflect the "additional experiments, investigations, and insightful explanations!"  Needless to say, score changes at this point are likely to be pivotal in deciding the paper's acceptance.

---

> ### Comment · Reviewer_WQNj · 2024-11-27
> **Thanks for the feedback, increased rating**
>
> Regarding Figure 21, I am happy this got sorted out.
>
> I agree with your priority assessments about the color bars. I am particularly interested in a color bar for the cases where $\gamma\neq 1$. You might also consider mentioning the name of the color map.
>
> **Taking the overall discussion and paper updates into account, I have increased my rating.** Thank you for your effort!
>
> ---
>
> **Some more comments about notation confusion**
>
> After looking at the code, I see that I misinterpreted the FVU equation (Eq. (3)). Is the following interpretation correct? The variance is computed over inputs (from the mini-batch?), token representations (from all tokens and layers), and features (with indices $1..d$). That is, given a set $D$ of representation vectors from multiple inputs, tokens and layers, it is the variance of the sample $\{\mathbf x[i] : i\in \{1..d\}, \mathbf x\in D\} \subset \mathbb R$, where $\mathbf x[i]$ is the $i$-the feature of the vector $\mathbf x$.
>
> What confused me was that (1) variance is usually not defined for random vectors (covariance is), (2) the function in the paper is not computed based on a single pair of vectors, but based on a set of vectors.
>
> For the camera-ready version of the paper, I suggest carefully checking the meaning of the symbols and making the notation more clear.

---

> > ### Author Response · Authors · 2024-12-03
> > **Response to Reviewer WQNj**
> >
> > Thank you for acknowledging the changes and updating your score! We apologize for the previous error in Figure 21.
> >
> > For the avoidance of doubt, we use the 'magma' colormap from Matplotlib, which is perceptually uniform and suitable for grayscale conversion (https://matplotlib.org/stable/users/explain/colors/colormaps.html#sequential). We will include the name of the colormap and a color bar for non-linearly mapped ($\gamma \neq 1$) figures in the camera-ready.
> >
> > Given an input tensor of shape (number of layers, batch size, sequence length, number of input features), we compute the variance over the final three dimensions, i.e., `torch.var(inputs, dim=(1, 2, 3))` in PyTorch. The number of layers, batch size, and sequence length form the mini-batch (or set $D$) of activation vectors, but we compute the variance for each layer individually because the activation vectors differ in scale (L223–226; see also Figure 4). We understand that our choice of notation is unclear, and we will make this clear in the camera-ready.

---

> ### Comment · Reviewer_WQNj · 2024-12-03
>
> Thank you! It seems that i had misread the `dim` argument – I apologize. This makes more sense.

---

### Official Review · Reviewer_g44V · 2024-11-03

**Soundness:** 4
**Presentation:** 4
**Contribution:** 3
**Rating:** 8
**Confidence:** 4

**Summary:**

The paper introduces and analyzes a multi-level sparse autoencoder (MLSAE), which is a sparse autoencoder (SAE) trained on latent representations of the residual stream at all layers. The work experiments with Pythia models on language modeling tasks. The paper further reinforces the technique by adding tuned-lens transformations, allowing analysis of the residual stream even if the information is encoded on different dimensions at different layers. The work then studies the behavior and similarity of representation at different layers in Transformers of varying sizes.
[After the discussion period: I have increased the score from 6 to 8, and increased the confidence from 3 to 4.]

**Strengths:**

The work experiments with Pythia models (Transformers) of different sizes, making the work highly reproducible. The experiments are sound and most of the doubts I had while reading were answered already in the text. While the paper doesn't offer revolutionary insights on its own, it is an excellent improvement to sparse autoencoders and may be very useful for mechanistic interpretability. The model also tests the relative performance of MLSAE at different expansion factors and sparsity levels and runs good ablations with "tuned lens". While I wondered about some design decisions, in particular "feature-stacked" MLSAE, when reading the paper, the discussion provided in Section 5 answered my potential questions before I even wrote them down. All in all, the writing is excellent, and the experiments look solid and reproducible. In general, I am leaning towards acceptance to the conference, and I am willing to adjust the score during the discussion.

**Weaknesses:**

1. The biggest weakness that I see is the lack of direct experimental comparison to previous techniques. I'd expect a paper to compare MLSAE and existing SAE techniques, similar to the comparison in Figure 5 of MLSAE w/ and w/o tuned lens. For example, I think it would be possible to show a version of Figure 1 (and almost all other figures, too) using previous SAE techniques, even without training on most of those layers. This would give the reader a qualitative understanding of whether MLSAE provided additional value.
2. Similarly, a quantitative measure (e.g., how much different methods agree on a similarity of different representations) could be made. In my eyes, the paper lacks a clear baseline to show improvement over.
3. The above is especially important as the paper on its own doesn't offer that many additional insights into Transformer - apart from the surprising result of isolated latent activations. The method also is quite simple.
4. I would also like to see an analysis of the method's weaknesses - for example, it seems like it should have a higher computational cost of training, but it is not clear what is the exact impact of that (I assume the impact is quite small).

**Questions:**

Please refer to the weaknesses section, specifically:
* regarding (1 and 2) - can you provide direct comparison to other methods or explain why such comparisons are absent?
* regarding (4)  what is the additional overhead when training/using MLSAE?

---

> ### Author Response · Authors · 2024-11-25
> **Response to Reviewer g44V**
>
> Thank you for your positive comments, acknowledging that our method "is an excellent improvement to sparse autoencoders and may be very useful for mechanistic interpretability" and "the writing is excellent, and the experiments look solid and reproducible."
>
> > The biggest weakness that I see is the lack of direct experimental comparison to previous techniques. I'd expect a paper to compare MLSAE and existing SAE techniques, similar to the comparison in Figure 5 of MLSAE w/ and w/o tuned lens. For example, I think it would be possible to show a version of Figure 1 (and almost all other figures, too) using previous SAE techniques, even without training on most of those layers. This would give the reader a qualitative understanding of whether MLSAE provided additional value.
>
> > Similarly, a quantitative measure (e.g., how much different methods agree on a similarity of different representations) could be made. In my eyes, the paper lacks a clear baseline to show improvement over.
>
> Thank you for calling attention to this. We agree with reviewers g44V and d1f6 that understanding whether MLSAEs provide additional value over standard, single-layer SAEs requires we make explicit comparisons between the two. We argue that the natural baseline to compare against is a standard SAE, trained on the activation vectors from a single layer of the residual stream, but applied to the activation vectors from every layer. Hence, we have now trained single-layer SAEs on each layer of Pythia-70m and 160m and evaluated their performance in terms of the FVU reconstruction error and the increase in the cross-entropy loss when we replace the input activations at a given layer by their SAE reconstruction. In the revised paper, we have included evaluation results for single-layer SAEs in Appendix B.4 and Figures 15–17. These figures show that an MLSAE performs similarly to a single-layer SAE when evaluated on the same layer as its training data but that an MLSAE performs much better than a single-layer SAE evaluated on a different layer. Importantly, the poor performance of single-layer SAEs when applied to activations from other layers means that, while it is possible to reproduce Figures 2 and 3 (for example) by applying a single-layer SAE to the activation vectors from every layer, the single-layer SAE may not faithfully represent the original LLM activations, so it is unclear whether the distributions of latent activations over layers can be considered meaningful.
>
> > The above is especially important as the paper on its own doesn't offer that many additional insights into Transformer - apart from the surprising result of isolated latent activations.
>
> Please see our global response above for our understanding of our mechanistic contributions.
>
> > The method also is quite simple.
>
> We believe the simplicity of the method is to its advantage. The success of existing SAE techniques is partly due to their simplicity, which has enabled researchers to apply them to varied problems and neural network architectures. We hope MLSAEs will have similar advantages (albeit limited to architectures with residual connections, like transformers).
>
> > I would also like to see an analysis of the method's weaknesses - for example, it seems like it should have a higher computational cost of training, but it is not clear what is the exact impact of that (I assume the impact is quite small).
>
> To the best of our knowledge, the strengths and weaknesses of our method are directly inherited from the underlying SAE. As for computational cost, an MLSAE is ultimately an SAE trained on different data (i.e., the activation vectors from every layer of the residual stream instead of a single layer). Hence, we expect the cost per activation vector encountered during training to be the same. Naturally, if you are training on 1 million tokens, then you have 1 million activation vectors for a single-layer SAE versus $L$ million for a multi-layer SAE (and $L$ layers). So, it is cheaper to train a single standard SAE on a given number of tokens than a single MLSAE. However, it is more common to train $L$ standard SAEs, one on each layer, at which point the computational cost becomes very similar to training a single MLSAE. In the revised paper, we briefly describe the computational cost of training MLSAEs at the end of Section 3.2.

---

> > ### Comment · Reviewer_g44V · 2024-12-02
> >
> > I want to thank the authors for their detailed replies, both to me and to other reviewers. Taking the rebuttal and discussion into consideration, when my concerns have been adequately addressed, I am increasing my score from 6 to 8, and likewise increasing my confidence from 3 to 4.

---

### Author Response · Authors · 2024-11-25
**Mechanistic insights**

A key question raised by most reviewers was the mechanistic insights into the behavior of transformers that our extensive experiments provide. We argue that a core subject of the paper is the phenomenon of representation drift, i.e., whether ‘concepts’ in the residual stream change their location in vector space as we step through layers of a transformer, and that this phenomenon is important to many forms of mechanistic analysis. For instance, direct logit attribution (Elhage et al., 2021; Wang et al., 2022) and ‘logit lens’ (nostalgebraist, 2020) are popular methods to analyze circuits and locate model behavior, but both methods implicitly assume that there is no representation drift, i.e., that we can interpret residual stream activations from different layers in the same way. Our work clarifies that this is not always the case (in part, by building on Belrose et al., 2023). Specifically, we find that many MLSAE latents become active at early layers but cease to remain active well before the last layer, and some latents seem to activate at a single layer only.

We believe that developing a better understanding of representation drift will likely help develop novel methods for mechanistic analysis. In this way, our paper provides important groundwork by demonstrating the lack of consistency between representations at different layers, and we hope that future work will build on this to develop more robust techniques. Finally, we agree with Reviewer d1f6 that identifying correspondences between “SAE latents across multiple layers is an important and understudied problem.” In particular, we are aware of two other submissions to ICLR 2025 that address this question:

* https://openreview.net/forum?id=MDvecs7EvO
* https://openreview.net/forum?id=NB8qn8iIW9

### References

Belrose et al., 2023: https://arxiv.org/abs/2303.08112

Elhage et al., 2021: https://transformer-circuits.pub/2021/framework/index.html

nostalgebraist, 2020: https://www.lesswrong.com/posts/AcKRB8wDpdaN6v6ru/interpreting-gpt-the-logit-lens

Wang et al., 2022: https://arxiv.org/abs/2211.00593

---

### Author Response · Authors · 2024-12-02
**Summary of discussion**

We thank the reviewers for their constructive and detailed comments.

The reviewers agreed that our work:
* "is an excellent improvement to sparse autoencoders and may be very useful for mechanistic interpretability” (g44V)
* “the writing is excellent, and the experiments look solid and reproducible” (g44V)
* “the paper is well written” and “the experimental evaluation is well done” (WQNj)
* "I believe your work has a great contribution to the community" (F8UN)
* "The author presents comprehensive experimental evidence to elucidate the statistical characteristics of representations at different layers" (F8UN)
* "an important and understudied problem, and the contribution of training a single SAE on activations from multiple layers is novel" ... "The paper is clear and understandable" (d1f6)

We are grateful for the productive and insightful discussion with all the reviewers during the discussion period.

---

### Meta-Review · Area_Chair_vKBA · 2024-12-23

**Metareview:**

This paper presents a solid extension of current SAE analysis techniques, specifically focusing on the multilayer activations within these models. Initially, some reviewers raised concerns about the absence of comparisons to established baseline methods. This omission made it difficult to definitively assess the novelty and improvements offered by the authors' approach. However, the authors responded promptly and comprehensively to this feedback during the rebuttal phase. They incorporated the suggested baseline comparisons, directly addressing the reviewers' concerns. Following the authors' revisions, all reviewers expressed their satisfaction with the paper's quality and contribution. They now confidently recommend acceptance of this work.

**Additional Comments On Reviewer Discussion:**

Main questions were around adding appropriate baselines for SAE analysis, which the authors did. Other concerns were around model size used for analysis, which the authors promised to address in the final version.

---

### Decision · Program_Chairs · 2025-01-22

Accept (Poster)